# Neuron-reactive KIR⁺CD8⁺ T cells display an encephalitogenic transcriptional program in autoimmune encephalitis

Sylvain Perriot [1,10], Samuel Jones [1,10], Raphaël Genolet [2,3], Amandine Mathias [1], Helen Lindsay[4], Sara Bobisse[2,3], Giovanni Di Liberto [5,6], Mathieu Canales[1], Lise Queiroz[2,3], Christophe Sauvage[2,3], Ingrid Wagner [5], Larise Oberholster[1], Marie Gimenez[1], Diane Bégarie[1], Stjepana Kovac[7], Virginie Desestret[8], Marie Théaudin [6], Caroline Pot [6], Heinz Wiendl [9], Jérôme Honnorat [8], Doron Merkler [5], Raphaël Gottardo [4], Alexandre Harari [2,3] & Renaud Du Pasquier [1,6] ✉

Autoreactive CD8⁺ T cells targeting neurons are the principal suspects in autoimmune encephalitis (AIE), but supporting data is still lacking. Here we identify neuron-reactive CD8⁺ T cells in a cohort of six healthy donors and one patient with anti-Ri encephalitis (Ri-AIE) by querying natural antigen presentation of neurons that are derived from human induced pluripotent stem cells. Single-cell RNA sequencing of ex vivo CD8⁺ T cells in an extended cohort of seven Ri-AIE patients and three aged-matched controls further reveal that these neuron-reactive CD8⁺ T cells correspond to cytotoxic KIR⁺CD8⁺ regulatory T cells. Intriguingly, KIR⁺CD8⁺ T cells from most Ri-AIE patients have reduced expression of KIR and the key regulatory transcription factor, Helios, encoded by the *IKZF2* gene; by contrast, these cells show activated TCR signaling and increased *TNF* and *IFNG* gene expression. Importantly, Ri-AIE-derived KIR⁺CD8⁺ T cells from blood also express higher levels of *TOX*, a gene associated with encephalitogenic potential, and is expressed in cytotoxic CD8⁺ T cells in the brain lesions of one Ri-AIE patient. Altogether, our data hints that dysregulated activity of neuron-reactive cytotoxic KIR⁺CD8⁺ T cells may contribute to Ri-AIE pathogenesis.

Autoimmune encephalitis (AIE) is a heterogenous group of neurological syndromes characterized by the presence of autoreactive antibodies targeting neuronal or glial antigens located in the intracellular compartment or cellular surface[1]. In the former case, it is suspected that, following an immunogenic event such as tumor development or infection, autoreactive CD8⁺ T cells from the periphery drive disease pathology through the infiltration of the central nervous system (CNS) and direct recognition of antigens presented by HLA class I molecules on neurons[2–4]. In this context, tumor- or pathogen-related antigens would activate peripheral CD8⁺ T cells and

neurological damage would arise without the necessity of an initial CNS-related insult[3]. As compared to AIE with surface-targeting antibodies, AIE with intracellular-targeting antibodies shows a higher brain infiltration of cytotoxic CD8⁺ T cells with a higher number of granzyme B⁺ T cells affixed to neurons and a correlation of those CD8⁺ T cells with neuronal damage[2,3,5]. Oligoclonal CD8⁺ T cell expansion has been reported in the blood, brain tissue, and dorsal root ganglia of patients with AIE[6–8]. Similarly, other paradigmatic neuroimmunological diseases such as Rasmussen encephalitis (RE) and Susac syndrome (SuS) also display oligoclonal CD8⁺ T cell

expansion, thus suggesting a common CD8+ T cell-driven pathology in these disorders[9,10].

While the phenotype and T cell repertoire of CD8+ T cells in RE and SuS have been vastly studied[9,10], similar studies with in-depth CD8+ T cell assessment are mostly lacking for AIE. Despite many efforts, neuron-reactive CD8+ T cells have not been consistently found in AIE patients. Circulating cdr2 (Yo)- and HuD-specific CD8+ T cells have been identified in paraneoplastic cerebellar degeneration and anti-Hu encephalitis patients, respectively[6,11–13]. Yet, other groups have also contested evidence for circulating CD8+ T cells against both these antigens[14–16].

In summary, evidence regarding the phenotype of allegedly autoreactive pathogenic CD8+ T cells in AIE is scarce and identification of neuron-reactive CD8+ T cells is still elusive[6–10]. One of the reasons for the limited results published so far is that previous studies were restricted to testing CD8+ T cell reactivity against the same antigens as the ones recognized by antibodies. Yet, there is no evidence that both the humoral and the cellular immune response should necessarily be directed against the very same protein. Additionally, whilst certain authors assessed a broad range of candidate epitopes by generating peptide libraries and transfecting antigen-presenting cells (APCs) with HLA molecules, these methods have never been applied to studying AIE and also have limitations[17]. In particular, these techniques require the pre-selection of a restricted number of antigens and HLA alleles. Furthermore, these techniques, coupled with subsequent conventional readouts, usually are not able to detect CD8+ T cell clonotypes that are present at a frequency lower than 0.01%[18].

Given the state of the art, we believe that there is a need to develop novel methodologies to address the limitations of current techniques. First, the search for neuron-reactive CD8+ T cells should be conducted in a fully human and, ideally, autologous setting. Second, the experimental setup must include the largest possible range of neuronal antigens to develop an unbiased approach. Third, this approach should be able to take into account protein isoforms or post-translational modifications that can affect antigen recognition by CD8+ T cells[19,20].

Here, to fulfill all these criteria, we develop a method to assess CD8+ T cell activation against a broad repertoire of antigens naturally produced and presented by autologous hiPSC-derived neurons. Then, given the strong rationale for the role of autoreactive CD8+ T cells in AIE, we explore the presence of neuron-reactive CD8+ T cells in a total of seven patients with Ri-AIE. We identify neuron-reactive CD8+ T cells in both control and diseased donors and uncover key dysregulated pathways specifically in CD8+ T cells from Ri-AIE patients. Overall, this study provides evidence for the implication of dysregulated auto-reactive cytotoxic CD8+ T cells in Ri-AIE and, most importantly, it presents a novel methodology that can be applied to other auto-immune diseases.

## Results

To assess the global CD8+ T cell response against neurons, we resort to the use of autologous human induced pluripotent stem cell (hiPSC)-derived neurons[21,22] as APCs. First, to validate this approach, we generated hiPSCs from six healthy donors (HDs) and differentiated these hiPSCs into neurons. We then assessed their capacity to trigger antigen-dependent CD8+ T cell activation. Upon stimulation with IFNγ and TNFα, we found that hiPSC-derived neurons display HLA class I molecules at their surface (Fig. 1a, b, Supplementary Fig. 1). We next assessed the capacity of hiPSC-derived neurons to activate CD8+ T cells in an antigen-dependent manner. For this, we performed an overnight

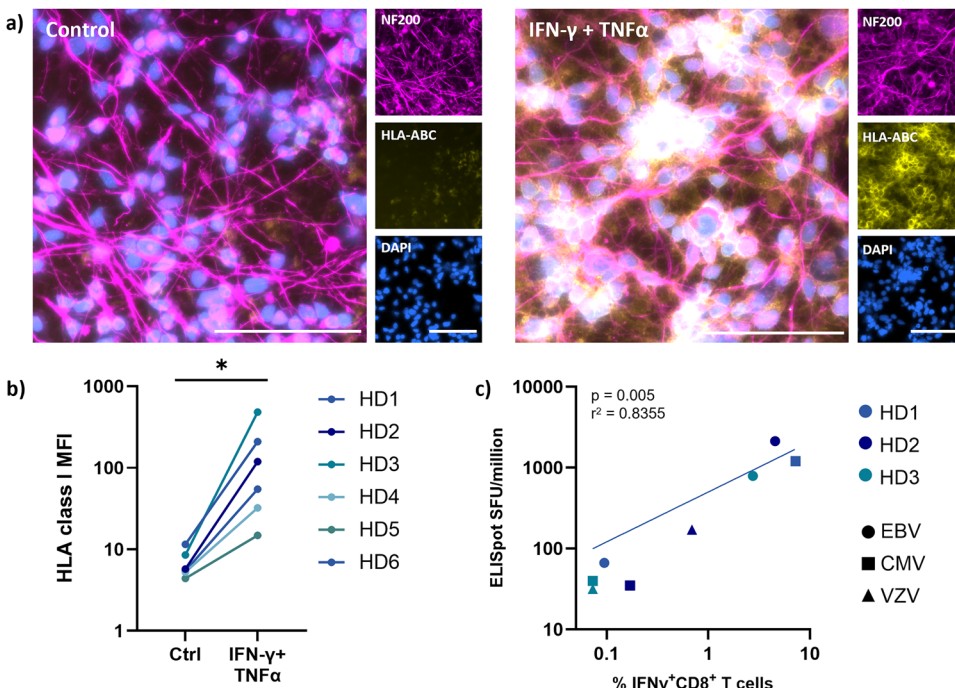

**Fig. 1 | Human iPSC-derived neurons trigger antigen-dependent CD8+ T cell activation. a** Representative fluorescence microscopy images of hiPSC-derived neurons from HD6, immunostained for NF-200 (magenta), HLA-ABC (yellow) and DAPI (blue), untreated (left panels, control) or treated with IFNγ and TNFα for 48 h (right panels, IFNγ + TNFα). Bar represents 100 μm. **b** Flow cytometry analysis of HLA class I expression (anti-HLA-ABC antibody (clone W6/32)) of hiPSC-derived neurons from 6 healthy donors (HD1-HD6), untreated (control) or treated with IFNγ and TNFα for 48 h (two-sided Wilcoxon test, p value = 0.0156). n = 6 biological

replicates (i.e., donors), in one experiment. **c** Correlation of the number of spot-forming units (SFU)/million PBMCs in an IFNγ ELISpot assay (y-axis) versus the percentage of IFNγ+CD8+ T cells as measured by IFNγ secretion assay (x-axis). In both assays, PBMC (ELISpot) or neurons (IFNγ secretion assay) were pulsed overnight with a pool of CD8+ T cell-restricted immunodominant viral epitopes from EBV, CMV and VZV. Correlations were assessed by running a two-sided Pearson correlation test. n = 9 biological replicates (3 peptide pools per 3 donors), in one experiment.

stimulation of ex vivo CD8[+] T cells with autologous neurons pulsed with CD8[+] T cell-restricted viral peptide pools (see Supplementary Tables 1 and 2). In this assay, we were able to elicit an IFNγ secretion by CD8[+] T cells that correlates with the secretion induced in total peripheral blood mononuclear cells (PBMCs) during a standard IFNγ ELISpot assay, demonstrating the capacity of neurons to trigger antigen-dependent activation of CD8[+] T cells (Fig. 1c).

As neuron-reactive CD8[+] T cell clonotypes are also present in healthy donors[23], we initially sought to demonstrate the feasibility of our method in a cohort of HDs. To identify neuron-reactive CD8[+] T cell clonotypes, we developed a coculture system between PBMCs and autologous hiPSC-derived neurons and monitored CD8[+] T cell proliferation after in vitro autologous coculture. We assessed the TCR-β chain repertoire of CD8[+] T cells after 14 days of culture and compared it to the ex vivo CD8[+] T cell repertoire (Fig. 2a). We first performed this assay using PBMCs from 6 HDs and performed conventional TCR-β chain repertoire metric assessments (Shannon entropy: a measure of TCR repertoire diversity; clonality: a measure of evenness of TCR chain distribution). We observed a global yet non-significant decrease in the clonality (average of 0.3 ex vivo vs 0.25 at 14 days, p value: 0.25) and of the Shannon entropy (average of 11.095 ex vivo vs 9.96 at 14 days, p value: 0.21) in the TCR repertoire after the culture (Fig. 2b).

We then analyzed the TCR β-chain sequences to identify reactive CD8[+] T cell clonotypes that would have been expanded upon coculture with autologous neurons. We considered a clonotype to be expanded in vitro if the TCR-β chain frequency presented with a ≥9-fold expansion as compared to ex vivo samples. After coculture, we observed several TCR-β chains whose frequencies were significantly increased after 14 days of culture (Fig. 2c). Focusing on TCR-β chains representing >0.5% of total repertoire at day 14, we could highlight a specific enrichment with a mean of 5.33 clonotypes per donor (range 1–12), representing from 0.5% to 7.2% of the total repertoire (Fig. 2c). Only one TCR-β chain was found at an ex vivo frequency >0.01% (HD1−TCR-β chain: 0.13%, asterisk in Fig. 2c). To ensure that these expanded clonotypes recognize neurons, we restimulated CD8[+] T cells with autologous neurons and sorted IFNγ[+]CD8[+] T cells in 96-well plates at one cell per well. We next amplified the mRNA sequences encoding the TCR-α and TCR-β chains in each well separately in order to reconstruct the TCRs of these cells. Using this approach, we selected 12 TCRs from HD4, which we cloned into Jurkat cells to assess their reactivity. We selected 6 clonotypes that we thought would recognize neurons based on their expansion kinetics in vitro, while the 6 other clonotypes were classified as non-expanded (did not display an exponential increase between day 7 and day 14 of coculture, Supplementary Table 3). When cultured with autologous neurons, 6/6 TCRs extracted from neuron-expanded CD8[+] T cells displayed a specific activation. The addition of an anti-HLA class I blocking antibody, preventing the TCR-HLA interaction, specifically abrogated this activation. Similarly, neurons that were not stimulated with IFNγ and TNFα, thus not presenting antigens through HLA class I molecules, did not elicit any TCR activation (Fig. 2d). Conversely, TCRs cloned from non-expanded (i.e., fold-increase <2 compared to ex vivo) CD8[+] T cells at day 14 did not activate in contact with autologous neurons, thus demonstrating that our technique efficiently expands neuron-reactive CD8[+] T cells (Fig. 2d).

Next, we looked for the presence of neuron-reactive CD8[+] T cells in a patient suffering from anti-Ri-AIE (Ri01), in which the autoantibodies target an intracellular antigen. As described for HDs, we generated hiPSC-derived neurons that upregulated HLA class I molecules upon IFNγ and TNFα exposure (43.4 fold change as measured by flow cytometry, Fig. 3a for fluorescence microscopy observations). We next performed our autologous PBMC-neuron coculture assay to assess whether we could identify neuron-reactive CD8[+] T cells in this patient. In sharp contrast with HDs (Fig. 2b), CD8[+] T cells from the Ri01 patient revealed the monoclonal expansion of one TCR-β chain, representing 58.81% of the total repertoire at day 14 (vs 6.47% ex vivo,

Fig. 3b). Interestingly, the TCR-β chain repertoire clonality increased from 0.34 ex vivo to 0.6 (1.76 fold increase) after coculture (Fig. 3b) while HD clonality displayed a decrease from a mean of 0.3 ex vivo to 0.25 (0.85 fold change) (Fig. 2b). To verify if the expanded TCR β-chain belonged to a TCR reactive for neurons, and since one concomitant TCR-α chain was also expanded at day 14, we cloned the paired TCR α- and β-chains into Jurkat cells and assessed their activation. Upon overnight coculture with neurons from Ri01, we could validate that this TCR recognizes neurons and that the antigen-dependent activation was blocked by the addition of an anti-HLA class I antibody (Fig. 3c). Additionally, the most prevalent yet non-expanded clonotype was used as a control and did not react against neurons (Fig. 3c).

Next, to further investigate the phenotype of CD8[+] cells in Ri-AIE, we performed single-cell RNA sequencing (scRNAseq) on sorted ex vivo circulating CD8[+] T cells from seven patients with Ri-AIE (including Ri01 patient for whom we had found the neuron-reactive CD8[+] T cell clonotype) as well as three sex- and age-matched donors without an autoimmune disease (AgD) (Fig. 4a, Supplementary Tables 4 and 5). Unsupervised clustering with all samples combined highlighted 16 different clusters (Fig. 4b) with different CD8[+] T cell populations according to conventional phenotyping markers[24]. Clusters 3–5 were enriched in naive CD8[+] T cells; clusters 0, 1, 13, and 14 in effector cytotoxic CD8[+] T cells; clusters 6 and 12 in activated CD8[+] T cells; and clusters 2, 10, and 15 in memory CD8[+] T cells (Supplementary Fig. 2). We observed a similar distribution of all samples across all clusters (Supplementary Fig. 3a, b). Furthermore, when comparing cluster repartition between AgD and Ri-AIE groups, we did not observe significant differences in CD8[+] T cell frequency per cluster, except for cluster 6, which was enriched in Ri-AIE patients (Supplementary Fig. 3c). Focusing on neuron-reactive CD8[+] T cells from Ri01 patient (Fig. 3), we noticed that 93.48% of the cells from this clonotype were present in cluster 1, the remaining minority of cells being localized in clusters 2, 8, 14, and 3 (Fig. 4c).

Given that almost all these identified neuron-reactive CD8[+] T cells were gathered in cluster 1, we turned to this unique cluster to further look for neuron-reactive CD8[+] T cells in the remaining six Ri-AIE patients and three AgDs. Interestingly, within this cluster, we found no difference in the TCR-β chain repertoire metrics (clonality and Shannon entropy) between the AgD and Ri-AIE groups (Supplementary Fig. 4a). Globally, cluster 1 was composed of 11,598 cells from 1184 individual clonotypes. We found 488 individual clonotypes from 5014 cells in all AgDs (average of 162.7 clonotypes/donor) versus 696 clonotypes from 6584 cells in Ri-AIE (average of 99.4 clonotypes/donor, Fig. 4d). Among them, 39.5% and 45.2% of these clonotypes were found expanded ex vivo (i.e. ≥2 cell per clonotype) in AgDs and Ri-AIE respectively, without significant differences between groups (Fig. 4d). Of note, expanded clonotypes represented 94.1% of all cells from cluster 1, congruent with an enrichment in effector CD8[+] T cells. To determine whether any of these ex vivo expanded clonotypes recognized neurons, we selected within cluster 1 several clonotypes per donor among those that displayed the strongest expansion. Clonotypes were selected according to the following criteria: (1) >10 CD8[+] T cells in cluster 1; and (2) >60% of cells from this clonotype present in cluster 1. We selected a total of 48 clonotypes, 27 from AgDs (49.1% of all expanded CD8[+] T cells ex vivo in cluster 1 from AgDs) and 21 from Ri-AIE (23.7% of all expanded CD8[+] T cells ex vivo in cluster 1 from Ri-AIE), fulfilling the above definition (Fig. 4d). We individually cloned these 48 TCRs into Jurkat cells (Supplementary Data 1) and assessed the luminescence induced by TCR activation after an overnight culture with partially HLA-matched hiPSC-derived neurons. On average, 4.66/6 HLA-class I alleles were matched for AgDs and 5.42/6 for Ri-AIE patients, with an average of 9 clonotypes/donor tested for AgDs and 3 clonotypes/donor for Ri-AIE (Supplementary Table 6). We observed an HLA-mediated TCR activation in 40.7% of AgD clonotypes and 42.9% of Ri-AIE clonotypes (no statistical difference) that was completely

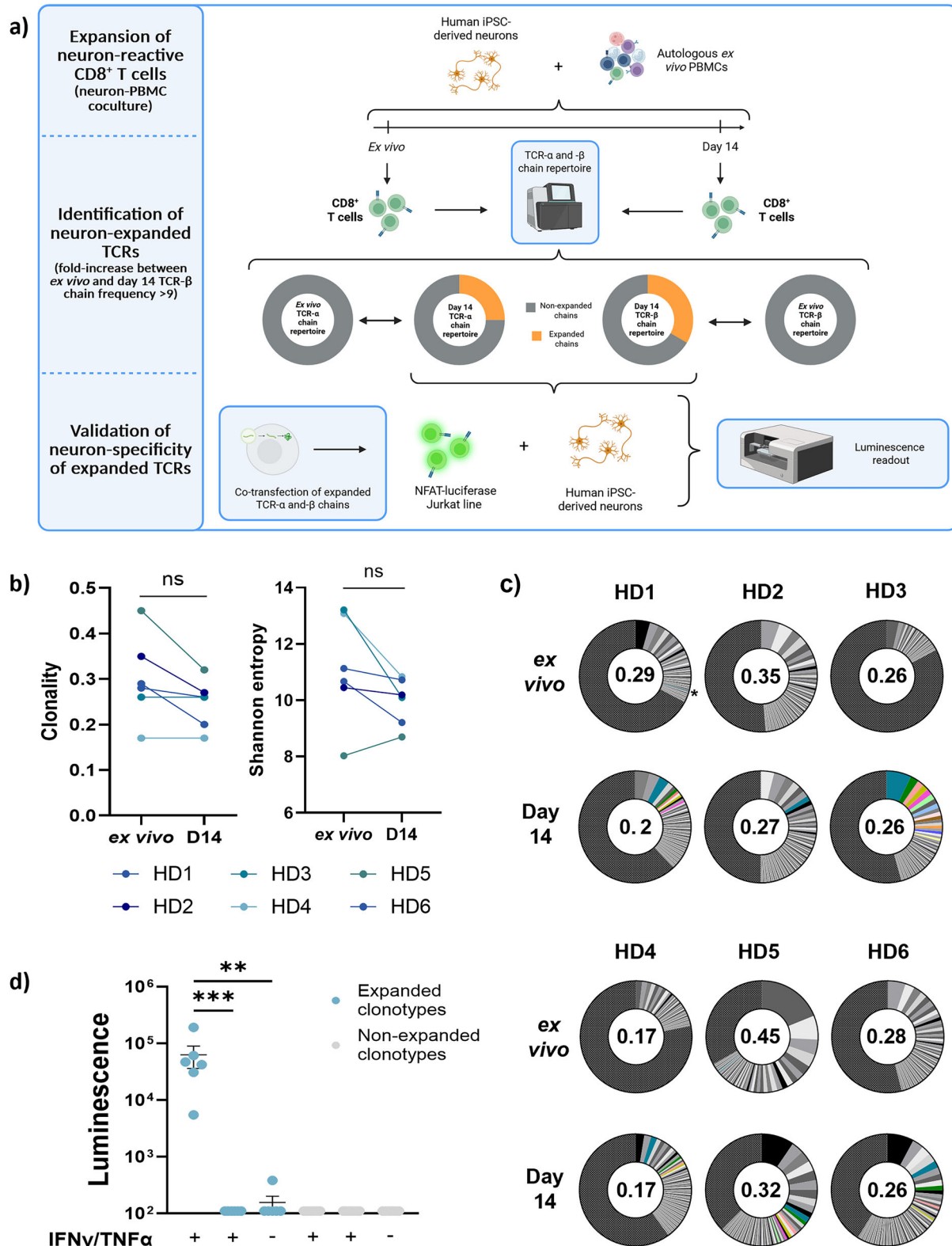

abolished in presence of a blocking anti-HLA-ABC antibody (Fig. 4e). We then assessed the ex vivo frequency of these neuron-reactive CD8[+] T cell clonotypes in AgD and Ri-AIE groups and observed no significant difference (Supplementary Fig. 4b). To investigate if these TCRs cross-reacted with peripheral antigens, we cocultured TCR-expressing Jurkat cells with PBMCs from the same donors as for the hiPSC-derived neurons. We found that 85% of the neuron-reactive TCRs also activated

upon culture with PBMCs, but with less intensity than neurons (Supplementary Fig. 4c). Altogether, these data demonstrate that auto-reactive CD8[+] T cells recognizing neurons are a common feature of the aged repertoire and not exclusive to Ri-AIE patients.

We next dived into the characterization of these neuron-reactive clonotypes to start unveiling the functionality of CD8[+] T cells from cluster 1. Among the most discriminant genes defining this cluster, we

**Fig. 2 | Human iPSC-derived neurons trigger the expansion of rare neuron-reactive CD8+ T cell clonotypes in healthy donors. a** Schematic representation of the experimental workflow for the identification of neuron-reactive CD8+ T cell clonotypes. First, a coculture of PBMCs with autologous neurons is performed for 14 days. Second, TCR-β chain repertoires at day 14 of coculture are compared to ex vivo TCR-β repertoires to identify expanded TCR-β chains. Third, paired TCR-α chains of expanded clonotypes are identified by scTCRseq, and both TCR-α and TCR-β chains are co-transfected into NFAT-luciferase reporter Jurkat CD8+ T cells, which are then cultured overnight with neurons. Luminescence values are then measured on a multimode microplate reader (see methods). Created in BioRender. Perriot, S. (2025) https://BioRender.com/duayzpo. **b** TCR-β chain repertoire clonality (top) and Shannon entropy (bottom) of CD8+ T cells ex vivo and after 14 days of neuron-PBMC coculture for all 6 HDs (two-sided Wilcoxon tests). n = 6 biological replicates (i.e., donors), performed once per donor across 2 independent experiments. **c** TCR-β chain repertoire analysis of the CD8+ T cells from 6 HD ex vivo and after 14 days of coculture. Each slice represents a unique TCR-β sequence. Slice size represents the frequency of each sequence in comparison with the total TCR-β chain sequences. Colored slices represent TCR-β chains that underwent a 9-fold expansion as compared to ex vivo frequencies, and that constitute >0.5% of total

TCR-β after 14 days of coculture. The asterisk indicates the only TCR-β representing more than 0.5% ex vivo (HD1). The value at the center of each pie chart represents the clonality of each TCR-β repertoire. All TCR-β sequences <1:1000 are grouped into one slice (black and grey patterned slice). n = 6 biological replicates (i.e., donors), performed once per donor across 2 independent experiments.
**d** Luminescence values of TCR-transfected NFAT-luciferase Jurkat cells after overnight culture with autologous neurons. Overnight culture was performed with Jurkat cells bearing TCRs that were identified as expanded in HD4 (blue) or prevalent TCR-β chains at day 14 that were non-expanded (grey). All luminescence values are adjusted to the background signal emitted by each respective TCR. For each clonotype, experimental conditions include neurons treated and non-treated with IFNγ + TNFα (thus HLA class I positive and negative, respectively) or with the addition of a blocking anti-HLA-ABC antibody (clone W6/32). Differences between resting vs stimulated conditions were assessed by a Kruskal−Wallis test: ns, not significant; **p < 0.01; ***p < 0.001. Expanded clonotypes, +IFNγ/TNFα vs +IFNγ/TNFα/anti-HLA: adj. p value = 0.0002. Expanded clonotypes, +IFNγ/TNFα vs -IFNγ/TNFα: adj. p value = 0.0021. n = 6 biological replicates (i.e., unique TCRs), this experiment was performed once. Data are presented as mean values ± SEM.

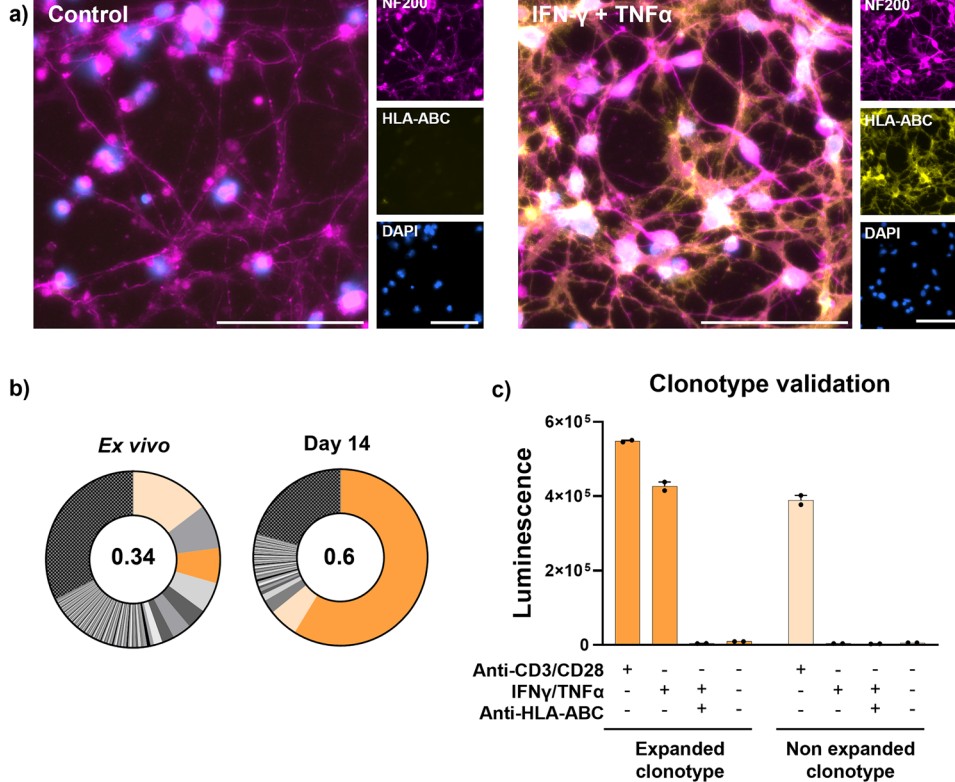

**Fig. 3 | Autologous hiPSC-derived neuron-reactive CD8+ T cells present with an important clonal expansion in a patient with Ri-AIE. a** Representative fluorescence microscopy images of hiPSC-derived neurons from Ri01 immunostained for NF-200 (magenta), HLA-ABC (yellow) and DAPI (blue), untreated (left panels, control) or treated with IFNγ and TNFα (right panels, IFNγ + TNFα) for 48 h. Bar represents 100 μm. **b** TCR-β chain repertoire analysis of the CD8+ T cells from Ri01 ex vivo (left) and after 14 days (right) of coculture with neurons. Each slice represents a unique TCR-β sequence, and the size of the slice represents the frequency of each sequence in comparison with total TCR-β sequences. Strongly expanded clone at D14 is displayed in dark orange with a prevalent yet non-expanded (i.e., control) clonotype in light orange. The value at the center of each pie chart represents the

clonality of each TCR-β chain repertoire. All TCR-β sequences <1:1000 are grouped into one slice (black and grey patterned slice). n = 1. **c** Luminescence values of TCR-transfected NFAT-luciferase Jurkat cells after overnight culture with neurons from Ri01. Experimental conditions include a transactivating anti-CD3/CD28 antibody without neurons, neurons treated or non-treated with IFNγ and TNFα or with the addition of a blocking anti-HLA-ABC antibody (clone W6/32). Jurkat cells were either transfected with the strongly expanded TCR identified in b) (dark orange) or with the most frequent yet non-expanded TCR at day 14 (light orange). Data are presented as mean values ± SEM. The expanded clonotype has been validated in n = 3 independent experiments.

found that killer cell immunoglobulin-like receptors (KIR), including *KIR2DL1*, *KIR2DL3*, *KIR2DL4*, *KIR3DL1*, *KIR3DL2* and *KIR3DL3*, ranked among the top 10 most upregulated genes along with the transcription factor *IKZF2* (Helios), and the killer cell lectin like receptors (KLR) *KLRC2* (NKG2C) and *KLRC3* (NKG2E) (Supplementary Data 2). These

data indicate that the CD8+ T cells from cluster 1 are enriched in KIR+CD8+ Treg cells. We further characterized the phenotype of these cells by assessing the expression of activation markers, costimulatory receptors, cytotoxic markers, immune regulation, checkpoint inhibitors, KIRs and a set of transcription factors (Fig. 4g). Overall, the

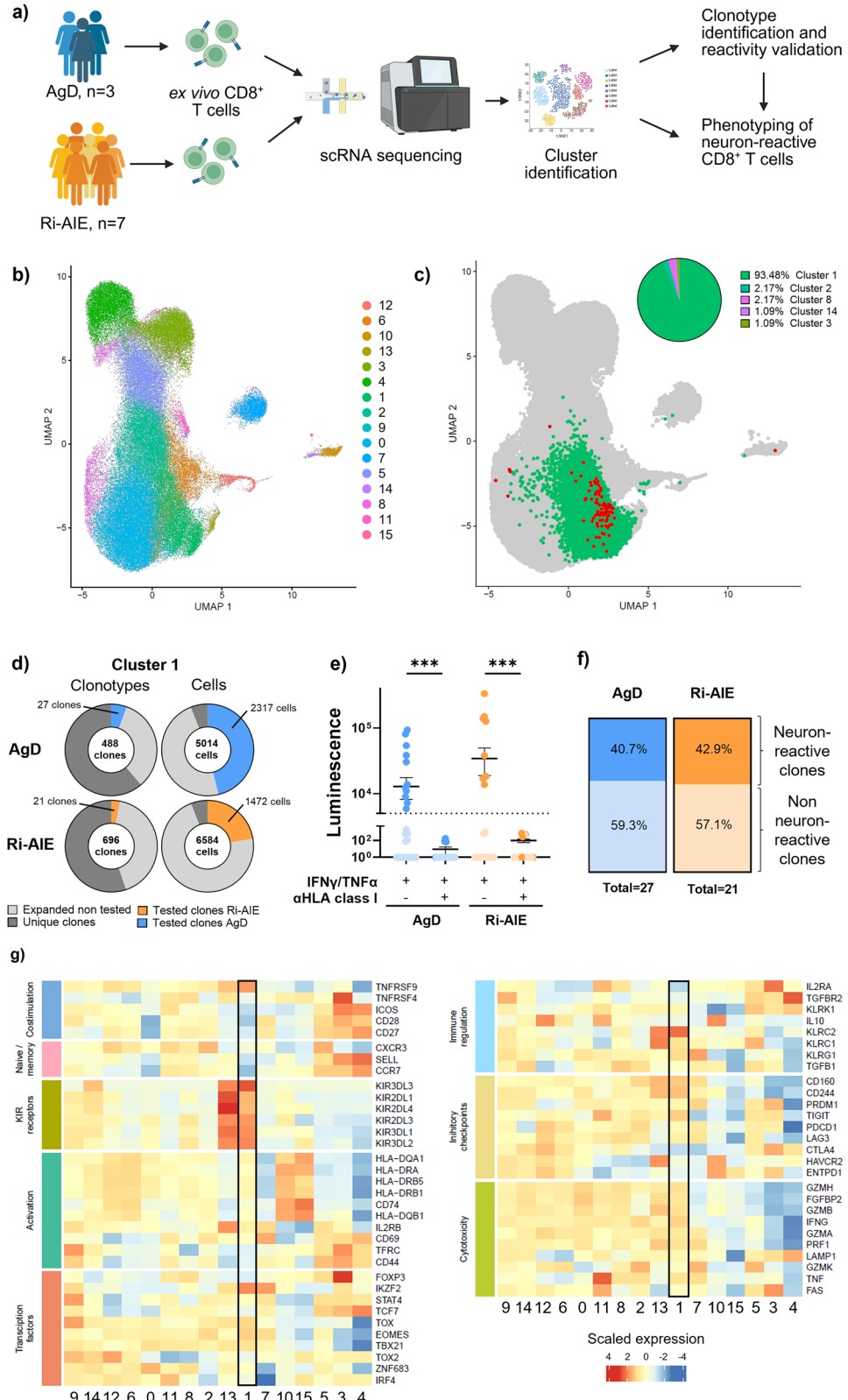

KIR$^+$CD8$^+$ Treg cells signature could be fully validated with cells from cluster 1 highly expressing not only the afore mentioned *KIR* family genes and the transcription factor *IKZF2* (Helios) but also additional transcription factors such as *TOX* and *EOMES*, an effector cytotoxic signature (*PRF1, GZMA, GZMB, IFNG, TNF*) as well as certain checkpoint inhibitors (*CD160, CD244, TIGIT*), concordant with *TOX* expression (Fig. 4g).

Interestingly, KIR$^+$CD8$^+$ Treg cells are known to be increased in aging, infection, cancer and autoimmunity[25–27]. Their primary function seems to be killing autoreactive CD4$^+$ T cells through classical cytotoxicity to prevent autoimmunity[25]. We thus hypothesized that neuron-reactive KIR$^+$CD8$^+$ Treg cells may display an altered phenotype in Ri-AIE, which would contribute to the development of CNS autoimmunity. We thus performed a differential gene expression analysis

**Fig. 4 | Neuron-reactive KIR⁺CD8⁺ T cells are a common feature in aged donors and Ri-AIE patients. a** Schematic representation of the experimental workflow for the identification of the neuron-reactive CD8⁺ T cell clonotypes and associated phenotypes. Single-cell RNA sequencing (scRNAseq) was performed on ex vivo circulating CD8⁺ T cells from seven patients with Ri-AIE (including Ri01) as well as three sex-and age-matched donors without an autoimmune disease (AgDs). Data were analyzed altogether by unsupervised clustering. Both neuron-reactivity (via TCR-transfection into NFAT-luciferase reporter Jurkat cells) and phenotype of selected clonotypes were further investigated in Ri-AIE vs AgD. Created in BioRender. Perriot, S. (2025) https://BioRender.com/nhxjm7k. **b** UMAP displaying unsupervised clustering of ex vivo CD8⁺ T cells from three AgDs and seven Ri-AIE patients assessed by scRNAseq, resulting in 16 different clusters. **c** Highlighting of cluster 1 (green) and CD8⁺ T cells presenting with the same TCR-β chain CDR3 amino acid sequence as the strongly expanded neuron-reactive clonotype from Ri01 (red). The pie chart on the top right highlights the cluster repartition of this clonotype, with each cluster represented in a different color and slices representing the percentage of cells belonging to each cluster from (**b**). **d** Pie charts representing the distribution of clonotypes (left) and cells (right) from AgDs (top) and Ri-AIE patients (bottom) present in cluster 1. The proportion of CD8⁺ T cells with one unique TCR is displayed in dark grey and those with ≥2 identical TCR in light grey (i.e., expanded ex vivo). Clonotypes were selected and tested for neuron-reactivity

if they presented with the following criteria: (1) a clonotype with ≥10 CD8⁺ T cells in cluster 1; (2) >60% of CD8⁺ T cells from this clonotype present in cluster 1. From CD8⁺ T cells fulfilling these criteria, we selected 27 from AgD (blue slice) and 21 from Ri-AIE (orange slice). **e** Dot plot representing luminescence values of TCR-transfected NFAT-luciferase Jurkat cells from AgD (blue) or Ri-AIE (orange) after overnight culture with HLA-enhanced neurons (matched for HLA haplotype). Horizontal dotted line represents positivity threshold established at a luminescence value of 5000 (neuron-reactive CD8⁺ T cell clonotypes are highlighted in darker shades of blue (AgD) or orange (Ri-AIE)). For both groups, a control condition with a blocking anti-HLA-ABC was included to ensure that the activation of Jurkat cells was TCR-mediated (two-sided Wilcoxon test, AgD $p$ value = 0.001, Ri AIE $p$ value = 0.0003 among neuron-reactive CD8⁺ T cells). $n$ = 27 for AgD and $n$ = 21 for Ri-AIE, neuron-reactive clonotypes were validated once per TCR across two independent experiments. Data are presented as mean values ± SEM. **f** Bar plots summarizing the proportion of neuron-reactive clones (blue) vs non-reactive clones (light blue) among AgDs (left) and Ri-AIE patients (orange vs light orange, right). **g** Heatmap displaying differentially expressed genes among all 16 clusters ($x$-axis). Genes displayed on $y$-axis are classified into functional categories conventionally used to classify CD8⁺ T cell phenotype. Upregulated genes are displayed in red with downregulated genes in blue.

at the single cell level, comparing the transcriptomic profile between neuron-reactive KIR⁺CD8⁺ Treg cells in cluster 1 from Ri-AIE patients and AgDs. We found 238 upregulated genes (logFC > 0.5, adj-$p$ value < 0.05) and 437 downregulated genes (logFC < −0.5, adj-$p$ value < 0.05; Fig. 5a, Supplementary Data 3). The top five most enriched KEGG pathways corresponded to antigen processing and presentation, Leishmaniasis, rheumatoid arthritis, Th17 cell differentiation, and human T-cell leukemia virus 1 infection (Fig. 5b). Because these KEGG pathways are associated with increased T cell activation and cytotoxicity, we selected a large panel of genes covering important aspects of CD8⁺ T cell functions, such as KIR-related markers, effector functions (cytotoxicity and cytokine production) as well as TCR activation and TCR inhibition, activation markers and pro-inflammatory signaling. First, neuron-reactive KIR⁺CD8⁺ Treg cells from Ri-AIE patients, as compared to AgDs, displayed a significant decrease in markers associated with Treg phenotype, in particular *IKZF2* (Helios), genes of the KLR family (*KLRC2, KLRC3, KLRK1*) and the inhibitory receptors *KIR2DL1* and *KIR2DL3*. Receptors from the KIR3DL subfamily tended to decrease as well (Fig. 5c). Interestingly, *KIR2DL4*, the only activator receptor of the KIR2DL family, displayed a trend of increase (1.77-fold increase). As KIRs act by modulating TCR downstream signaling, we sought to assess if this global downregulation of inhibitory receptors was associated with TCR activation. Consistent with this decrease of inhibitory receptors, neuron-reactive KIR⁺CD8⁺ T cells from Ri-AIE displayed a strong increase in genes associated with TCR activation (*FOS, FOSB, JUN, JUND, RELB, EGR1, EGR2*; Fig. 5d) and a global trend to a decrease in genes active in TCR signaling inhibition, *INPP5D* (SHIP1) and *CBLB* being significantly downregulated (Fig. 5e). We next investigated if this observation translated into an increase in activation markers and effector function. As expected, most of the activation markers studied displayed a significant increase as compared to neuron-reactive KIR⁺CD8⁺ T cells from AgDs (*HLA-DQA1, HLA-DQB1, HLA-DRA, HLA-DRB1, CD44, CD69, CD74, ICAM1*; Fig. 5f). Interestingly, we observed a significant increase in genes involved in pro-inflammatory cytokine and chemokine production (*IFNG, TNF, CCL3, CCL4, CSF1*). Additionally, these cells displayed a decrease in *GZMH* while they retained an important expression of other cytotoxic genes (*GZMK, LAMP1, NKG7*) (Fig. 5g). Overall, the functional profile of neuron-reactive KIR⁺CD8⁺ Treg cells appears to be disrupted in Ri-AIE with a decrease of markers associated with the regulatory function but a strong increase in TCR signaling, overall activation and cytokine production, suggesting a shift toward a more activated and potentially pathogenic phenotype. In order to assess interindividual variability, we

aggregated all cells for each donor to conduct a pseudobulk analysis at the donor level. This analysis showed consistent patterns across multiple donors and evidenced trends confirming the analysis performed at the single-cell level (Supplementary Figs. 5 and 6).

Additionally, we compared CD8⁺ T cells from Ri01 during disease and during clinical remission (5 months later, when the patient was treated by mycophenolate mofetil and corticosteroids) to explore if the neuron-reactive CD8⁺ T cell clonotypes identified during disease would present with phenotypical differences during remission. Due to the limited number of cells analyzed, we found very few dysregulated genes between the two timepoints (Supplementary Fig. 7a). Specifically, *FOSB* and *JUND* were the two most dysregulated genes, both significantly increased at the disease timepoint. Taking into account the most dysregulated genes in Ri-AIE, we could observe that the cells from Ri01 at disease peak mostly segregated together from the cells from Ri01 at remission suggesting global changes in the gene expression profile despite a low individual significance (Supplementary Fig. 7b). Strikingly, most gene expression changes in Ri-AIE at disease peak versus remission mirrored the dysregulation found in neuron-reactive CD8⁺ T cells from Ri-AIE versus AgDs (Supplementary Fig. 7c). These data suggest a global reverting of the pathogenic phenotype of neuron-reactive CD8⁺ T cells concomitant with clinical improvement.

Last, since TOX was one of the key transcription factors associated with cluster 1 (Fig. 4g, Supplementary Data 2), and as it plays a major role in T cell exhaustion and triggering an encephalitogenic program in CD8⁺ T cells, we looked at the expression of genes controlled by TOX[28]. Some genes known to be upregulated by TOX (*ZFP36L1, NFKBIA*) were also upregulated in neuron-reactive KIR⁺CD8⁺ T cells from Ri-AIE, while the genes downregulated by TOX mostly displayed a trend to decrease (only *ID2* was significant) (Fig. 5h). These data again suggest that TOX is active in this subset of neuron-reactive CD8⁺ T cells.

Having demonstrated the effector and activated phenotype of neuron-reactive KIR⁺CD8⁺ T cells in Ri-AIE as compared to AgD, we addressed one critical question: are these cells present in the CNS? Indeed, to be able to trigger symptoms in Ri-AIE patients, these cells would have to be present in the brains of patients. As we were able to obtain autopsy tissue for one RI-AIE patient from our cohort (Ri02), we looked for the presence of cytotoxic CD8⁺ T cells in the lesions of this patient as compared to age-matched controls (Fig. 6a, b). We observed a massive CD8⁺ T cell infiltration (mean of 7 CD8⁺ T cells/mm²) in Ri02 patient as compared to five age-matched controls (mean of 1 CD8⁺ T cell/mm²; Fig. 6c). Supporting a role for TOX in inducing an

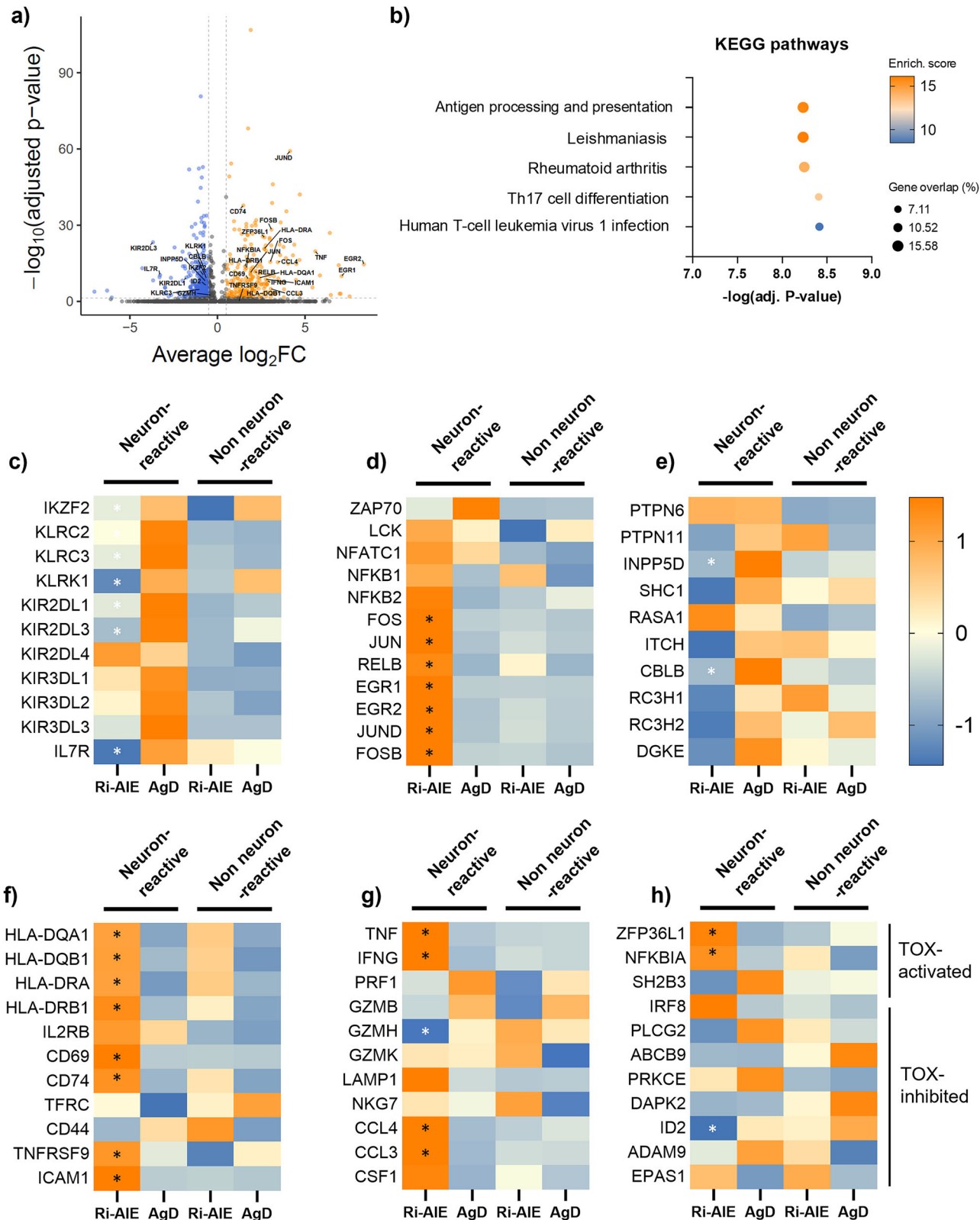

encephalitogenic program in CD8+ T cells[28], we found that 58.8% of CD8+ T cells in these brain lesions were positive for TOX (Fig. 6d). Additionally, 82.2% of CD8+ T cells in the lesions were GzmB+ and affixed to neurons (Fig. 6e).

Overall, our results demonstrate that while blood-circulating neuron-reactive KIR+CD8+ T cells are commonly present in aged

individuals, those of Ri-AIE patients have lost their regulatory phenotype, are more activated and are present in the brain. Importantly, this disease-associated phenotype was partially reverted in CD8+ T cells from one of these patients taken during clinical remission. Altogether, these findings assert the pathogenesis of autoreactive CD8+ T cells in Ri-AIE.

**Fig. 5 | Neuron-reactive KIR⁺CD8⁺ T cells from cluster 1 in Ri-AIE patients are strongly activated and express high levels of pro-inflammatory cytokines.** **a** Volcano plot highlighting all significantly upregulated (orange) and down-regulated genes (blue) between neuron-reactive KIR⁺CD8⁺ T cells from Ri-AIE vs AgD. Vertical ticked lines represent the fold-change threshold set at an average log₂FC(0.5) for upregulated or log₂FC(−0.5) for downregulated genes. Horizontal ticked line represents the adjusted *p* value threshold established at 0.05. All significantly differentially expressed genes from c−h are annotated on the plot. **b** Dot plot of KEGG pathway enrichment analysis highlighting the 5 most enriched pathways according to adjusted *p* value, in neuron-reactive KIR⁺CD8⁺ T cells from Ri

AIE vs AgD. The enrichment score is scaled from blue to orange (strongest score), with each dot reflecting the percentage of gene overlap relative to each pathway. **c−h** Heatmaps displaying selected genes involved in key CD8⁺ T cell functions such as immune regulation (**c**), TCR activation (**d**) or inhibition (**e**), activation markers (**f**), pro-inflammatory signaling (**g**) or TOX-controlled genes (**h**). Highly expressed genes are represented in orange and low-expressed genes in blue. A star is displayed in the column for neuron-reactive KIR⁺CD8⁺ T cells from Ri-AIE when genes from this group were significantly differentially expressed as compared to neuron-reactive KIR⁺CD8⁺ T cells from AgDs.

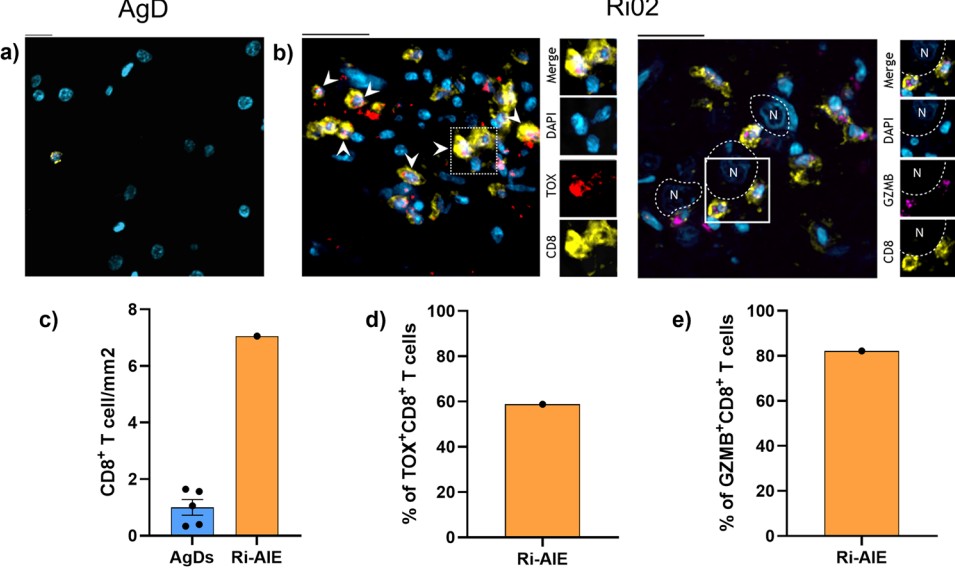

**Fig. 6 | Cytotoxic TOX⁺CD8⁺ T cells infiltrate the brain in Ri-AIE. a** Representative immunostaining of brain tissue from one aged-matched control. Brain tissue was stained for CD8 (yellow) and DAPI (blue). Bar length represents 20 μm. **b** Immunostaining of a brain lesion in autopsy tissue from Ri02. Left panels: Brain tissue was stained for TOX (red), CD8 (yellow) and nuclei (blue). Bar length represents 20 μm. Right panels: Brain tissue was stained for GZMB (magenta), CD8

(yellow), and nuclei (blue). Bar length represents 25 μm. **c** Quantification (cells/mm²) of brain-infiltrating CD8⁺ T cells in Ri02 (orange) vs aged-matched donors (*n* = 5, blue). Each dot represents one individual sample. Bar height represents the mean ± SEM. **d** Quantification of the proportion of TOX⁺CD8⁺ T cells among total brain-infiltrating CD8⁺ T cells for Ri02. **e** Quantification of the proportion of GZMB⁺CD8⁺ T cells among total brain-infiltrating CD8⁺ T cells for Ri02.

## Discussion

Here, we report on an unbiased novel method to identify neuron-autoreactive CD8⁺ T cells in a human-based and autologous system. The antigen-presenting cells used in this technique are the cells naturally producing the antigens in the body and, therefore, can present the broadest range of antigens with the right cell-type-associated post-translational modifications and splicing. Indeed, such antigen modifications have been demonstrated to strongly influence T cell antigenic recognition in cancer[29], and it has been known for a long time that citrullination, for example, plays a role in antigen recognition in autoimmune diseases[30]. In addition, by using autologous neurons, we can simultaneously screen for T cell recognition against all relevant HLA alleles for the studied cohort.

Applying this method to first screen for the presence of neuron-reactive CD8⁺ T cells in young HDs (median age 36 years old), we could uncover that six out of six donors demonstrated expansion of specific T cell clonotypes upon coculture with neurons. Further validation of individual TCRs showed that expanded CD8⁺ T cells in these conditions are indeed neuron-reactive. Although it has been previously shown that certain HDs may harbor hypocretin neuron, astrocyte or oligodendrocyte-reactive CD8⁺ T cells[23,31,32], these results highlight that the presence of circulating neuron-reactive CD8⁺ T cells is a highly common feature of the human TCR repertoire. Our data demonstrate that they are most strikingly present at very low frequencies in all young HDs studied (mostly below 0.01%, Fig. 2c). Of note, identification of these neuron-reactive CD8⁺ T cells

in young HDs was possible only thanks to the very high sensitivity of our assay which depends only on the number of T cells used as input and demonstrated sensitivity superior to 0.001% (with an input of 10⁵ CD8⁺ T cells). In comparison, conventional methods are mostly limited to frequencies above 0.01%[18]. As such, the methodology described here holds great promise to study the CD8⁺ T cell repertoire in patients with autoimmune diseases but also in HDs, thus providing the scientific community with the right tools to address fundamental questions on the development of autoreactive CD8⁺ T cells and how they can go rogue in autoimmunity.

In the second part of this study, we focused our analysis on the identification of neuron-reactive CD8⁺ T cells in a larger cohort of three aged control donors without autoimmune disease (AgDs, median age 70.5 years old) and seven aged-matched Ri-AIE patients (median age 69 years old). Again, neuron-reactive CD8⁺ T cells were identified in all donors, but contrasting with the results from younger donors, ex vivo neuron-reactive CD8⁺ T cells were more abundant in Ri-AIE and AgDs (ranging from 0.01% to 6.47%, Supplementary Fig. 4b). Of note, there was no difference in terms of frequencies of neuron-reactive CD8⁺ T cells between AgDs and Ri-AIE. Due to the partially HLA-matched system we used to assess TCR reactivity, we cannot rule out that certain neuron-reactive CD8⁺ T cell clonotypes may have been missed due to incomplete HLA matching. Nevertheless, the similar proportion of neuron-reactive CD8⁺ T cell clonotypes between AgDs and Ri-AIE patients mitigates this concern.

Resorting to single-cell transcriptomic analysis coupled with TCR sequencing, we uncovered that these autoreactive clonotypes belonged to a subset of CD8+ T cells expressing the KIR family and the transcription factor Helios (IKZF2). These markers are characteristic of KIR+CD8+ Treg cells. They have been shown to be expanded in aging, infection, cancer and various autoimmune diseases[25,33,34]. KIR expression on CD8+ T cells plays a role in long-term survival as well as controlling TCR activation to prevent excessive and deleterious effector functions[35]. While the antigen specificity of KIR+CD8+ Treg cells has not been fully elucidated, a few studies reported that they most likely recognize autoantigens through classical TCR-HLA class I (either A, B, C) or HLA-E interactions[25,36,37]. Interestingly, recent studies demonstrated that one of the key targets of KIR+CD8+ Treg cells is CD4+ T cells, in particular autoreactive ones, which contribute to preventing autoimmunity[25]. In cancer, KIR+CD8+ Treg cells infiltrate tumors and are associated with a bad prognosis since they are acting on other immune cells to dampen the immune response against tumors[27]. These seminal studies contributed to identifying KIR+CD8+ T cells as a novel Treg subset acting through cytotoxic effector functions[25,27].

Interestingly, KIR+CD8+ T cells can be increased in autoimmune diseases, raising the question of their actual regulatory capacities in these diseases[38]. Actually, phenotypic differences of KIR+CD8+ Treg cells in the healthy state versus autoimmune diseases have not been thoroughly described. In the present study, we show that KIR+CD8+ T cells can recognize neurons, notably in the context of Ri-AIE, a disease in which autoreactive CD8+ T cells are suspected to be the main pathogenic effectors[2]. Strikingly, we find in the single cell analysis that the expression of KIR genes (in particular the KIR2DL family), IKZF2 and genes involved in inhibiting TCR downstream signaling are significantly decreased in neuron-reactive CD8+ T cells from the Ri-AIE group as compared to aged-matched controls. Concomitantly, these cells display strong TCR signaling coupled with increased expression of activation markers and cytokine production, thus demonstrating increased effector functions.

In line with our findings, Li et al. reported increased expression of activation and cytotoxic markers in low KIR-expressing CD8+ T cells vs high KIR-expressing CD8+ T cells[25]. Of importance, this study explored the role of KIR+CD8+ Treg cells in various autoimmune conditions, including multiple sclerosis (MS), the most frequent neurological autoimmune disease. Strikingly, KIR+CD8+ T cells in MS were characterized by the increase of a specific subset displaying strong IFN signaling[25]. Mirroring this finding, we found a specific gene set related to IFN signaling to be increased in neuron-reactive KIR+CD8+ T cells from Ri-AIE. Of note, all Ri-AIE patients but one suffered from cancer as an underlying cause of AIE. Yet, it is suspected that the autoreactive CD8+ T cells responsible for AIE are primed within tumors before targeting the CNS. It has been demonstrated that tumors from patients suffering from AIE with anti-Hu antibodies, another CD8+ T cell-mediated AIE, were characterized by increased IFNγ signaling, mirroring our finding of high IFN signaling in neuron-reactive KIR+CD8+ T cells in Ri-AIE[39]. Altogether, our results raise the question whether the microenvironment of some tumors may provide high TCR stimulation together with IFN signaling to infiltrating KIR+CD8+ Treg cells. The latter cells would, in turn, downregulate KIRs, overruling inhibitory control, thus gaining the ability to cause AIE by directly recognizing neurons through their TCR.

Interestingly, our results demonstrate that these KIR+CD8+ T cells produce TCRs able to recognize both neurons and immune cells, albeit at a lower magnitude. Interestingly, a similar cross-reactivity has previously been demonstrated for CD4+ T cells in the context of MS, with some CD4+ T cells recognizing the RASGRP2 antigen expressed by B cells and cortical neurons[40]. Given that KIR+CD8+ Treg cells are known to recognize autoreactive CD4+ T cells through their TCR, by analogy, we suggest that dysregulated CD8+ T cells in AIE may not only recognize immune cells, but also neuronal antigens[25]. However, to be more assertive, the specificity of these autoreactive KIR+CD8+ T cells would have to be studied more in depth, especially because 15% of the neuron-reactive TCRs we identify recognize only neurons and not PBMCs.

We also found that TOX was one of the transcription factors associated with the KIR+CD8+ T cell cluster in our dataset. Interestingly, TOX is mostly known in the context of chronic infection and cancer, in which it plays the dual role of enhancing long-term survival of CD8+ T cells but also driving T cell hyperresponsiveness through exhaustion[41,42]. This functionality is extremely similar to the KIR family receptors, suggesting that both pathways may be linked. Most importantly, TOX is a key regulator of the encephalitogenic program in autoreactive CD8+ T cells, as demonstrated in experimental models of brain autoimmunity[28]. Furthermore, TOX+CD8+ T cells have been found in the brain parenchyma of patients with AIE, as well as in MS, thus suggesting a strong role of this factor in neurological autoimmune disorders[28,43]. As a matter of fact, the neuron-reactive KIR+CD8+ T cells we identified in the blood of Ri-AIE patients share some markers with encephalitogenic autoreactive CD8+ T cells in these models. In particular, we find dysregulation of downstream genes directly regulated by TOX. Moreover, we also demonstrate that TOX+CD8+ T cells are found in the lesions of a Ri-AIE patient (RiO2) for whom we had autopsy brain tissue, thus confirming previous findings pertaining to TOX association with AIE.

Taken together, our findings suggest that in AIE, autoreactive KIR+CD8+ T cells most likely shift from their original purpose of containing autoimmunity towards infiltration of the brain, thus contributing to the pathogenesis of neurological autoimmune disorders. Functional studies would be warranted to further explore how these autoreactive KIR+CD8+ T cells directly cause the damage seen in Ri-AIE lesions. We demonstrate that neuron-reactive clonotypes downregulate inhibitory markers and display a specific gene signature of highly activated cytotoxic CD8+ T cells together with a signature reminiscent of the TOX-induced encephalitogenic program. Of note, we observed some degree of variability among Ri-AIE patients with alterations in the aforementioned pathways being more marked in some patients than others, thus warranting further confirmation in a larger cohort of patients. Indeed, while statistical analysis at the single cell level revealed an interesting set of genes significantly dysregulated, statistical analysis at the patient level in a pseudobulk/mixed-model analysis (3 AgDs vs 5 Ri-AIE) did not identify a significant difference, most likely because of limited sample size and uneven cell numbers.

Here, we focused on the study of KIR+CD8+ T cells due to the initial identification of a neuron-reactive clonotype belonging to this cell population in Ri01. However, we cannot exclude that other CD8+ T cell populations may also play a role in Ri-AIE, for instance cluster 6 was significantly enriched in Ri-AIE patients (Supplementary Fig. 3c). Finally, our coculture method based on using hiPSC-derived as APCs has the potential to advance the understanding of pathogenic CD8+ T cells in neurological autoimmune diseases, paving the way for significant insights into this intricate field, and suggests that hiPSC-derived somatic cells could be used as antigen-presenting cells to study a large variety of other autoimmune diseases.

## Methods
### Study population
Blood samples from six healthy donors (HDs), three AgDs and seven patients with anti-Ri-AIE were used in this study (Supplementary Table 4). Blood samples were collected across three centers [Laboratory of Neuroimmunology (Lausanne University Hospital and University of Lausanne); Department of Neurology with Institute of Translational Neurology (University of Münster, Germany); and the French Reference Center on Paraneoplastic Neurological Syndrome (Université Claude-Bernard, Lyon, France)]. For brain tissue, one Ri-AIE

patient (Ri02, age 70) and 5 age-matched controls (Department of Pathology and Immunology, University of Geneva) (F/M ratio: 2/3, median age 70) were included. All donors gave their written informed consent according to regulations established by the responsible ethics committees (Lausanne, Project COOLIN'BRAIN, CER-VD 2018-01622; Geneva, Project n° 2017-01737; Ethics committee of the University of Münster registration nos. 2013-682-b-S and 2016-053-f-S; Institutional review board of the Hospices Civils de Lyon). PBMCs from each donor were obtained by standard Ficoll-Paque (Sigma-Aldrich) gradient centrifugation, and cells were liquid nitrogen-frozen in fetal bovine serum (FBS, Biowest) and DMSO (1:10, Sigma-Aldrich)[44]. PBMCs from all AIE patients were taken during diagnostic workup when symptoms were present and patients were untreated, with an additional time-point at remission also taken for Ri01. For HLA typing, genomic DNA was extracted from PBMCs, and HLA genes were amplified by poly-merase chain reaction (PCR). Nextera adapters were added by tag-mentation, and the resulting libraries were sequenced on the MiniSeq instrument (Illumina). Sequencing data were then analyzed with the Assign TruSight HLA v.2.1 software provided by CareDx[45].

### Generation and characterization of human-induced pluripotent stem cell (hiPSC)-derived neurons and precursors

Human iPSCs were generated as part of the Lausanne biobank (BB_007_BBH-NI). For all six HDs and one Ri-AIE patient (Ri01), hiPSCs were reprogrammed from CD71[+] cells isolated from PBMCs and passed all standard quality controls required (microbiology, pluripotency, differentiation capacity, genomic integrity, episome clearance)[46,47]. Human iPSCs were differentiated into neural precursor cells (NPCs)[46,48], and mature neurons were then obtained through trans-duction of NPCs with an NGN2 lentiviral vector as described by Ho et al.[22] and used in previous publications[49,50]. Human iPSC-derived neurons were characterized by immunohistochemistry as follows. First, they were plated and differentiated in polyornithine (1:5, Sigma-Aldrich)/laminine (1:500, Sigma-Aldrich)-coated tissue culture-treated 24-well plates and treated with IFNγ (1 ng/ml, Miltenyi Biotec) and TNFα (1 ng/ml, R&D Systems) for 48 h to induce HLA class I enhance-ment (see Human-iPSC-derived neuron preparation section and Sup-plementary Table 7). Neurons were washed once with 500 μL of cold PBS and fixed with PBS + paraformaldehyde (PFA) (4:100, Electron Microscopy Sciences) for 10 min at 4 °C. Wells were then washed with blocking buffer (PBS + Normal goat serum (5:100, Jackson ImmunoR-esearch) + 0.1% Triton X-100 (Sigma-Aldrich)). Primary antibodies including rabbit IgG anti-NF200 (1:200, Sigma Aldrich, polyclonal), chicken IgY anti-MAP2 (1:200, Abcam, polyclonal) and mouse IgG2a anti-HLA-A, -B, -C (1:500, AffinityImmuno, clone W6/32) were incu-bated in 250ul blocking buffer with neurons overnight at 4 °C. After extensive washing steps, secondary antibodies including anti-rabbit IgG (H + L) AF546 (Invitrogen), anti-chicken IgY (H + L) AF546 (Invi-trogen) and anti-mouse IgG (H + L) AF488 (Invitrogen) were added for 30 min at room temperature. Counterstaining with DAPI (1:500, Sigma-Aldrich) was also performed at this stage (Supplementary Table 8). Then, after additional washing steps, images were acquired using a Leica DMi8 microscope and post-processed with Leica LAS X software V5.2.

### Human iPSC-derived neuron preparation (HLA enhancement and peptide pulsation)

Neurons were plated in polyornithine (1:5, Sigma-Aldrich)/laminine (1:500, Sigma-Aldrich)-coated tissue culture-treated 48-well plates (Corning) at a density of 100,000 cells/cm². To induce HLA class I upregulation, the cell medium was replaced with fresh medium sup-plemented with IFNγ (1 ng/ml, Miltenyi Biotec) and TNFα (1 ng/ml, R&D Systems) for 48 hours. Some experiments required the pulsation of CNS cells with reconstituted viral peptide pools. To this end, CNS cells were pulsed with CD8[+] T cell-restricted peptide pools from either EBV,

CMV or VZV (1 μg/ml for each peptide pool, JPT Peptide Technologies) four hours prior to culture with CD8[+] T cells. EBV and CMV CD8[+] T cell-restricted peptide pools were reconstituted from respectively 29 and 45 individual immunogenic peptides (JPT Peptide Technologies) (Supplementary Tables 1 and 2). VZV peptide pools were acquired from a commercially available mix of 63 individual peptides (PepMix VZV, IE63, JPT Peptide Technologies). Importantly, prior to overnight incubation with CD8[+] T cells, hiPSC-derived neurons were carefully washed four times with cell medium to remove any residual cytokines (i.e., IFNγ or TNFα) or viral peptides.

### Neuron-ex vivo CD8[+] T cell overnight culture

Ex vivo PBMC were thawed and rested overnight in a serum-free T cell expansion medium (SFM, Thermo Fisher Scientific). After resting, CD8[+] T cells were isolated by magnetic-associated cell sorting (MACS) (see MACS section below), counted and cultured with IFNγ/TNFα-treated neurons (±peptide pulsation) (see hiPSC-derived neuron preparation section and Supplementary Table 7) at a ratio of 1:1 (i.e., 100'000 CD8[+] T cells for 100'000 neurons per well). Neurons and autologous CD8[+] T cells were cultured together overnight, and an IFNγ secretion assay was performed. Regarding IFNγ ELISpot, steps were performed as per manufacturer instructions[44].

### Neuron-PBMC coculture

For all 6 HDs and Ri01, autologous PBMCs were thawed and rested overnight in SFM. Rested PBMCs were counted, resuspended at 4 million cells/ml and then cultured with IFNγ/TNFα-treated neurons (see hiPSC-derived neuron preparation section and Supplementary Table 7) at a density of 1.2 million cells/cm² with IL-2 (1000 UI/ml, Miltenyi Biotec) and a CD28 agonist antibody (5ug/ml, BD Bios-ciences, clone CD28.2) for a total of 14 days. In total, 300 μl of fresh SFM and IL-2 (1000UI/ml) were added at day 2 and then renewed at day 5. At day 7, PBMCs from the coculture were harvested, counted and resuspended at 4 million cells/ml and then re-cultured with fresh IFNγ/TNFα-treated neurons. In total, 300 μl of fresh SFM and IL-2 (1000 UI/ml) were added at day 10 and then renewed at day 12. Bulk TCR-α and -β chain repertoire sequencing was performed directly ex vivo as well as at days 7 and 14 of coculture (see Bulk TCR-α and TCR-β chain repertoire sequencing and analysis section and Supple-mentary Table 7). In some cases, an IFNγ secretion assay was per-formed at day 14 of coculture and subsequent fluorescence-associated cell sorting (FACS) was performed to isolate CD3[+]CD8[+]IFNγ[+] fractions (see Flow cytometry section). Of note, due to the restricted number of PBMCs, neuron-PBMC cocultures were not performed for any of the AgDs nor for the remaining six Ri-AIE patients (Ri02-Ri07). Instead, in-depth scRNA seq was prioritized over hiPSC reprogramming and subsequent coculture experiments. Indeed, for these patients, PBMC yield was insufficient to: (1) generate hiPSC-derived neurons, (2) perform the autologous cocultures.

### Magnetic-associated cell sorting (MACS)

Isolation of untouched CD8[+] T cells from total PBMCs was performed using a two-step human CD8[+] T cell isolation kit (Miltenyi Biotec). All steps were performed following the manufacturer's instructions. After MACS, additional purity checks were performed by flow cytometry (see Flow cytometry section and Supplementary Tables 7 and 8).

### IFNγ secretion assay

This assay was performed in two separate experiments: (1) Overnight culture of ex vivo CD8[+] T cells with IFNγ/TNFα-treated neurons (±peptide pulsation); (2) overnight culture of CD8[+] T cells isolated after 14 days of neuron-PBMC coculture with IFNγ/TNFα-treated neurons. Experimental conditions included IFNγ/TNFα-treated neurons or the addition of a blocking anti-HLA-A, -B, or -C antibody (W6/32, Affinity Immuno).

IFNγ production of CD8$^+$ T cells was measured by adapting a commercially available IFNγ secretion assay kit (Miltenyi Biotec). Briefly, MACS-sorted CD8$^+$ T cells were stained with an IFNγ Catch Reagent prior to overnight culture with neurons. Additionally, an IFNγ Detection Antibody (1:100) was added to the neuron-CD8$^+$ T cell culture prior to overnight incubation. If not otherwise specified, all other steps and reagent concentrations were applied following the manufacturer's instructions.

## Flow cytometry

For HLA class I assessment, neurons were detached using TrypLE (Gibco) and washed with phosphate buffer saline (PBS) and FBS (2:100). Cells were then stained with a pan HLA-A, -B, -C antibody (2:100, Santa Cruz Biotechnology, clone W6/32), washed again with PBS + 2% FBS and fixed in PBS + PFA (4:100, Electron Microscopy Sciences).

CD8$^+$ T cell staining was performed the day after the overnight IFNγ secretion assay or for post-MACS purity checks. Cells in suspension were harvested and washed with PBS and 2% FBS. Anti-CD8a (2:100, BD Biosciences, clone RPA-T8) and anti-CD3 (2:100, BD Biosciences, clone SK7) antibodies were then added, cells were washed again with PBS + 2% FBS and fixed in PBS + 4% PFA. Viability of all cells was assessed using an aqua fluorescent reactive amine dye (4:1000, Life Technologies) (Supplementary Table 8). Surface marker expression was assessed using an LSR II flow cytometer (BD Biosciences) and BD FACSDiva Software V. Analysis was carried on FlowJo software V11.

When single-cell TCR sequencing was necessary (see "Single-cell TCR-α and TCR-β chain sequencing" section), after an overnight IFN-γ secretion assay, CD8+ T cells were sorted by fluorescence-associated cell sorting (FACS) into IFNγ+CD3+CD8+ T populations at 1 cell/well. For this, CD8$^+$ T cells were stained and processed as described above, and sorting was performed using a FACSAria III cytometer (BD Biosciences). No PBS-PFA fixation steps were performed in this case.

## Bulk TCR-α and TCR-β chain repertoire sequencing and analysis

MACS-isolated CD8$^+$ T cells were suspended in 300 µl Lysis/Binding Buffer (Thermo Fisher Scientific), and bulk TCR sequencing analyses were performed[51]. mRNA was isolated by magnetic beads and amplified by in vitro transcription using the Superscript III enzyme with primers specific for TCR gene families. A 5′ adapter was added by multiplex reverse transcription, and TCRs were amplified using one primer in the adapter and one in the constant region. Libraries were sequenced on an Illumina instrument, and TCR sequences were processed using an ad hoc *Perl* script. The Shannon Entropy was calculated as follows

$$-\sum_{i=1}^{n} F_i * log2(F_i) \qquad (1)$$

Where $n$ is the total number of clonotypes and $F$ the clonotype frequency. Clonality, refers to 1-Pielou index, was calculated as follows

$$1 - \left( \frac{-\sum_{i=1}^{n} F_i * log10(F_i)}{log10(n)} \right) \qquad (2)$$

Where $n$ is the total number of clonotypes and F the clonotype frequency.

A clonotype was considered expanded if its TCR-β chain frequency presented with a >9 fold-increase between day 14 of coculture and ex vivo. When TCR-β chains were undetectable ex vivo, the fold-increase was calculated by assigning a maximal theoretical frequency to the ex vivo TCR-β chain values (i.e., 1/n° of ex vivo CD8$^+$ T cells used for bulk TCR-β chain sequencing).

## Single-cell TCR-α and TCR-β chain sequencing (scTCRseq)

For the pairing of the corresponding TCR-α chain, scTCRseq on CD8$^+$ T cells at day 14 was performed. For this, IFNγ+CD3+CD8+ T cells were sorted by FACS at 1 cell/well into 96-well PCR plates containing 15 µL of DNase/RNase-free distilled water (Invitrogen) with 0.2% Triton X-100 (Sigma-Aldrich) and an RNase inhibitor (2 U/µL) (Enzymatics). TCR-α and -β chains were amplified using OneStep RT-PCR kit (Qiagen) according to the manufacturer's recommendations, with the following modification: a collection covering all V segments and two primers designed in alpha and beta constant regions were used for the amplification. Then, amplicons were purified using AmpureXP beads (Beckman Coulter) according to the manufacturer's instructions. A second amplification was performed with the Phusion Hot Start DNA Polymerase (NEB). A forward primer, designed in the adapter sequence (added during the RT-PCR) and two reverse nested primers, designed in the constant regions, were used for the amplification. The PCR mix was composed of 1 µL of purified product, 0.4 µL of 10 mM dNTP mix (Promega), 0.4 µL of primers mix *alpha*, 0.4 µL of primers mix *beta*, 2 µL of buffer and 0.2 µL of polymerase and 5.6 primers mix $H_2O$. TCR-α and -β chains were amplified with the following PCR cycles: 98 °C for 4′, 25× (98 °C for 10″, 55 °C for 30″, 72 °C for 30″), 72 °C for 2′. The PCR product was purified by adding 1 µL of ExoSAP-IT (Affymetrix) and incubating at 37 °C for 15′ and then at 85 °C for 15′. The third PCR was performed with the Phusion hot Start (NEB) to add the Illumina adapter and index. A mix containing 1 µL of 10 mM dNTP (Promega), 5 µL of 0.25 µM NexteraXT primer index, 3 µL of buffer, 0.2 µL of enzyme and 5.8 µL of deionized water was prepared and added directly to the purified PCR product. The second amplification was performed as follows: 98 °C for 4′, 15× (98 °C for 10″, 55 °C for 30″ 72 °C for 30″), 72 °C for 2′. After purification on AmpureXP beads (Beckman Coulter), libraries were sequenced on Illumina instruments and sequences extracted using an ad hoc Perl script. Both chains were amplified, then transfected into reporter cell lines as described below (see TCR-α and -β chain amplification, cloning and validation section and Supplementary Table 7).

## TCR-α and -β chain amplification, cloning and validation

For all the HDs tested and Ri01, single TCR-α and -β chains were amplified from residual bulk RNA material[51]. Briefly, two primers were designed in the CDR3 of each TCR to specifically amplify the V and the J regions. The two amplicons were combined by fusion PCR, and the constant region was added by a second fusion PCR.

For TCR chains from the three AgDs and Ri02-Ri07 AIE patients, fully human codon-optimized DNA sequences were synthesized at GeneArt (Thermo Fisher Scientific) and served as templates for in vitro transcription (IVT) and polyadenylation of RNA molecules as per the manufacturer's instructions (HIScribe T7 ARCA mRNA kit (with tailing), NEB), followed by co-transfection into recipient T cells.

To validate antigen reactivity, TCR-α/β pairs were cloned into a recipient Jurkat cell line (TCR/CD3 Jurkat-luc cells (NFAT), Promega, in-house stably transduced with human CD8$^{α/β}$ and TCR$^{α/β}$ CRISPR-KO)[52]. mRNA encoding for the TCR chains was generated using the HiScribe T7 ARCA mRNA Kit as per the manufacturer's instructions. Jurkat cells were electroporated using the Neon electroporation system (Thermo Fisher Scientific) with the following parameters: 1,325 V, 10 ms, 3 pulses. Electroporated Jurkat cells were cultured with IFNγ/TNFα-treated hiPSC-derived neurons at a ratio of 1:1 (i.e., 100,000 Jurkat for 100,000 neurons) in a 48-well plate in 200 µL SFM. All conditions were run in duplicates. After overnight incubation, 50,000 Jurkat cells were transferred into opaque tissue culture-treated 96-well plates (Corning) and the assay was performed using the Bio-Glo Luciferase Assay System (Promega). Additional controls include mock-(transfection with nuclease-free water) transfected Jurkat cells and culture with IFNγ/TNFα-treated neurons or in the presence of a blocking anti-HLA-ABC antibody (W6/32, AffinityImmuno). Luminescence was measured with

a Multimode Microplate Reader (BioTek Synergy, Gen 5 software). As a positive control for TCR activation, Jurkat cells were cultured in the presence of TransAct (i.e., an anti-CD3/CD28 activating compound., Miltenyi).

The testing of TCR-transfected NFAT Jurkat cells was performed against neurons from the donor in which the TCR was identified (HD4 and Ri01) or against HLA-matched donors when we did not have hiPSC-derived neurons available (Ri02-Ri07). In these cases, the TCR-transfected NFAT Jurkat cells were tested against hiPSC-derived neurons from other donors available in our biobank with matched HLA alleles (Supplementary Table 6).

## Single-cell RNA and VDJ sequencing

Single-cell RNA and VDJ sequencing were performed on ex vivo CD8$^+$ T cells from all three AgDs and all seven Ri-AIE donors. Of note, for Ri01, two samples were sequenced, one at disease peak and one during remission 5 months later (under treatment with mycophenolate mofetil) resulting in a total of eleven assessed samples. CD8$^+$ T cells positively sorted by MACS were assessed for viability by AO/PI staining and counted with a Luna-FX7 Automated cell counter (Logos Biosystems).

A Chromium Next GEM Chip N (10× genomics) was loaded with the appropriate number of cells, and the sequencing libraries were prepared with the Chromium Next GEM Single Cell 5′ HT Reagent Kits v2 dual index following the manufacturer's recommendations. Briefly, an emulsion encapsulating single cells, reverse transcription reagents, and cell barcoding oligonucleotides was generated. After the actual reverse transcription step, the emulsion was broken, and double-stranded cDNA was generated and amplified in a bulk reaction. This cDNA was fragmented, a P7 sequencing adaptor ligated, and a 5′ gene expression library was generated by PCR amplification. For V(D)J sequencing, a similar approach was followed except that 2 steps of PCR-based V(D)J target enrichment were performed prior to fragmentation.

Libraries were quantified by a fluorimetric method, and their quality was assessed on a Fragment Analyzer (Agilent Technologies). Sequencing was performed on Illumina NovaSeq 6000 v1.5 flow cells for 28-10-10-90 cycles (read1 - index i7 - index i5 - read2) with 1% PhiX spike in. Sequencing data were demultiplexed using the bcl2fastq2 Conversion Software (v. 2.20, Illumina) and primary data analysis performed with the Cell Ranger Gene Expression pipeline (version 7.1.0, 10× Genomics). Mapping of VDJ sequences to the gene expression library resulted in an average total of 97.69% mapped TCR-β chains (AgD1: 97.9%, AgD2: 97.7%, AgD3: 94.7%, Ri01_dis: 98%, Ri01_5m: 97.9%, Ri02: 97.3%, Ri03: 98.6%, Ri04: 98.3%, Ri05: 97.8%, Ri06: 99.6%, Ri07: 96.8%) and 63.27% TCR-α chains (AgD1: 83.9%, AgD2: 73.4%, AgD3: 67.5%, Ri01_dis: 69.2%, Ri01_5m: 60.7%, Ri02: 56.2%, Ri03: 85.8%, Ri04: 88.3%, Ri05: 49.3%, Ri06: 42.3%, Ri07: 19.4%).

## CD8$^+$ T cell clustering analysis and cell subtype annotation

Ambient RNA contamination was removed from the CellRanger counts using Cellbender v0.3.0[53], with the number of cells detected per sample from the initial CellRanger analysis used as the "expected-cells" parameter and 25,000 as the "total droplets included" parameter. TCR sequences were collated with cell barcodes using scRepertoire v2.3.2, quantifying all productive TCR chains. Filtered counts generated by Cellbender were further filtered to select cells with a single TCR-β chain sequence in the VDJ reads, a maximum of 10% mitochondrial reads and at least 100 genes expressed; and genes expressed in at least 10 cells; 109,478 cells (4824–13,569 per donor) remained after quality filtering. These cells were integrated and clustered using Seurat's atomic sketch integration[54] with 5000 cells used for sketching, 20 neighbours considered when selecting anchors, reciprocal PCA as the integration method, and default values for all other parameters.

UMAPs were run using the sketch integrated data with the first 30 PCA components and projected onto the remaining cells.

## Histology

For immunofluorescence staining, after antigen retrieval (Sodium Citrate pH6, 30 min) and blocking of unspecific binding (PBS and FBS (1:10)), PFA-fixed sections were incubated with primary antibodies (mouse IgG1 anti-CD8a (1:50, Abcam, clone C8/144B), rat anti-TOX (Thermo Fisher Scientific, clone TXRX10), mouse IgG2a anti-GZMB (1:20, Monosan, clone GrB-7). To amplify the signals of TOX, bound antibodies were visualized with appropriate species-specific Cy5-conjugated secondary antibodies for anti-rat tyramide signal amplification (TSA). Nuclei were stained with DAPI (Sigma-Aldrich) (Supplementary Table 8). Immunostained sections were scanned using Pannoramic 250 FLASH II (3DHISTECH) Digital Slide Scanner with objective magnification of 20×. Positive signals were quantified by a blinded experimenter using Pannoramic Viewer software (3DHISTECH) and an image analysis ruleset based on Visiopharm. For representative images, white balance was adjusted, and contrast was linearly enhanced using the tools levels, curves, brightness, and contrast in Adobe Photoshop CC. Image processing was applied uniformly across all images within a given dataset.

## Graphical representation

Schematic representations were created with BioRender.com. Graphical representations of HLA class I assessments, IFNy secretion assays, single-cell and bulk TCR sequencing, TCR validation experiments and KEGG analysis were developed using GraphPad Prism 9 software (Version 9.1.0). For RNA-seq visualization (heatmaps, UMAPs, dotplots, PCAs, volcano plots and barplots) and custom clonotype quantification, we used R version 4.4.0 (https://www.R-project.org/) and the following R packages (DOI: 10.32614/CRAN.package.argparse), circlize v. 0.4.16[55], ComplexHeatmap v. 2.20.0[56,57], ggrepel v. 0.9.6 (10.32614/CRAN.package.ggrepel), hdf5r v. 1.3.12 (DOI: 10.32614/CRAN.package.hdf5r), janitor v. 2.2.1 (DOI: 10.32614/CRAN.package.janitor), Matrix v. 1.7.3 (DOI: 10.32614/CRAN.package.Matrix), patchwork v. 1.3.0 (DOI: 10.32614/CRAN.package.patchwork), rhdf5 v. 2.48.0 (DOI: 10.18129/B9.bioc.rhdf5), scales v. 1.3.0 (DOI: 10.32614/CRAN.package.scales), scRepertoire v. 2.0.7[58], Seurat v. 5.2.1[54,59–62], SeuratDisk v. 0.0.0.9021 (https://github.com/mojaveazure/seurat-disk), tidyverse v. 2.0.0 (DOI: 10.21105/joss.01686) and GraphPad Prism® 9 software (Version 9.1.0). Pseudobulked data was used for heatmap visualizations.

## Statistical analysis

Statistical analyses were performed using GraphPad Prism 9 software (Version 9.1.0) and R version 4.4.0 (https://www.R-project.org/). Paired non-parametric Wilcoxon tests were performed to compare HLA class I expression (untreated vs IFNγ + TNFα) and ex vivo and day 14 TCR-β chain metrics (clonality and Shannon entropy). Unpaired Mann–Whitney tests were performed to compare ex vivo TCR-β chain metrics and frequencies between AgDs, as well as luminescence values between these two groups. P values < 0.05 were considered significant. A Pearson correlation test was performed to compare the number of SFU from the IFNγ ELISpot and the percentage of IFNγ$^+$CD8$^+$ T cells from IFNγ secretion assay. For scRNAseq, differential expression analyses were performed at the cell level using Seurat's FindMarkers method using the default Wilcoxon rank-sum test after removal of genes encoding the TCR-α and -β chains. For sample-level analyses, we used Seurat's Aggregate Expression to calculate pseudobulk counts and DESeq2 (doi:10.1186/s13059-014-0550-8) for testing for differential expression. In both cases, tests were run via Seurat's FindMarkers function, with an average log-fold change cutoff of 0.5 and a Benjamini-Hochberg false discovery rate of 0.05 used as significance cutoffs in cell-level analyses (Supplementary Data 2 and 3). For dot

plots, we used DESeq2's fpm function directly to calculate fragments per million for pseudobulk counts, to normalize for differences in fragment counts between samples.

## Reporting summary

Further information on research design is available in the Nature Portfolio Reporting Summary linked to this article.

## Data availability

The scRNA/VDJseq data generated in this study have been deposited in the Gene Expression Omnibus (GEO) database under accession code GSE263666. Additional data can be made available upon request to the corresponding author. All the rest of the data are included in the Supplementary Information or available from the authors, as are the unique reagents used in this Article. The raw numbers for charts and graphs are available in the Source Data file whenever possible. Source data are provided with this paper.

## Code availability

Code for the Seurat analyses is available at https://github.com/bdsc-tds/Perriot_Jones_2025 (10.5281/zenodo.16812576).

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

## Acknowledgements

The authors thank L. Arlettaz (ICH, Sion CH), G. Le Goff (CHUV, Lausanne CH), E. Schuman (UKM, Munster DE), A-L. Pinto, E. Peter and G. Picard (CHU, Lyon, FR) for their contributions in enrolling the patients and providing blood samples and demographic/clinical information. S.J. is supported by the Swiss National Science Foundation (323630_214544). D.M. is supported by the Swiss National Science Foundation (310030_215050 and 310030B_201271) and the ERC (865026). R.D.P. is supported by a generous donor advised by Carigest SA, and a part of this study was supported by the Swiss National Science Foundation (320030-179531). J.H. and V.D. are supported by a public grant overseen by the French Agence Nationale de la Recherche (ANR) as part of the second Investissements d'Avenir program (ANR-18-RHUS-0012) to collect anti-Ri PBMC and are members of the European Reference Network RITA.

## Author contributions

S.P. conceived the study and experiments, performed experiments, analyzed data, discussed results, drafted the paper and figures, and revised the manuscript. S.J. conceived experiments, performed experiments, analyzed data, discussed results, drafted the paper and figures, and revised the manuscript. R.Ge. helped conceive experiments, performed experiments, analyzed data, discussed results, and revised the manuscript. A.M. discussed results, revised the manuscript. H.L. analyzed data, drafted figures, and revised the manuscript. S.B. helped conceive experiments, analyzed data, discussed results, and revised the manuscript. G.D.L. performed experiments, analyzed data, discussed results, and reviewed the manuscript. M.C. performed experiments. L.Q. performed experiments, analyzed data. C.S. performed experiments, analyzed data. I.W. performed experiments, analyzed data. L.O. performed experiments. M.G. performed experiments. D.B. performed experiments. S.K. provided samples and feedback on the study, revised the manuscript. V.D. provided samples and feedback on the study, revised the manuscript. M.T. provided feedback on the study, revised the manuscript. C.P. discussed results, provided feedback on the study, and revised the manuscript. H.W. provided samples and feedback on the study, revised the manuscript. J.H. provided samples and feedback on the study, revised the manuscript. D.M. discussed results, provided feedback on the study, revised the manuscript, and provided funding. R.Go. discussed results and revised the manuscript. A.H. discussed results, provided feedback on the study, revised the manuscript, and provided funding. R.D.P. conceived the study, analyzed data, discussed results, revised the manuscript, and provided funding. R.Ge. and A.M. contributed equally to this work.

## Competing interests

S.P., S.J., A.M., H.L., G.D.L., M.C., L.Q., C.S., I.W., L.O., M.G., D.B., S.K., V.D., M.T., C.P., H.W., J.H., D.M., R.D.P. have nothing to disclose related to this work. R.Ge. has patents in technologies related to TCR repertoire analysis. S.B. and A.H. have patents in technologies related to T cell expansion and engineering for T cell therapy, none related to this work. R.Go. has received consulting income from Takeda, Sanofi, Arcellx (all payments to the institution) and declares ownership in Ozette Technologies, none related to this work.

## Additional information

[1]Laboratory of Neuroimmunology, Neuroscience Research Centre, Department of Clinical Neurosciences, Lausanne University Hospital and University of Lausanne, Lausanne, Switzerland. [2]Ludwig Institute for Cancer Research, Lausanne Branch, Department of Oncology, Lausanne University Hospital and University of Lausanne, Agora Cancer Research Center, Lausanne, Switzerland. [3]Center for Cell Therapy, Lausanne University Hospital and University of Lausanne-Ludwig Institute for Cancer Research, Lausanne, Switzerland. [4]Biomedical Data Science Center, University of Lausanne and Lausanne University Hospital, Swiss Institute of Bioinformatics, Lausanne, Switzerland. [5]Department of Pathology and Immunology, Division of Clinical pathology, University and University Hospitals of Geneva, Geneva, Switzerland. [6]Service of Neurology, Department of Clinical Neurosciences, University Hospital of Lausanne and Lausanne University Hospital, Lausanne, Switzerland. [7]Department of Neurology with Institute of Translational Neurology, University Hospital Münster, Münster, Germany. [8]French Reference Centre for Paraneoplastic Neurological Syndromes, Hospices Civils de Lyon; MeLis Institute, SynatAc Team, Inserm U1314/ UMR CNRS5284, Lyon, France. [9]Department of Neurology and Neurophysiology, University Medical Center, Freiburg, Germany. [10]These authors contributed equally: Sylvain Perriot, Samuel Jones. ✉e-mail: renaud.du-pasquier@chuv.ch

