## [Transparent Peer Review file · Nature Communications]

Neuron-reactive KIR+CD8+ T cells display an encephalitogenic transcriptional program in autoimmune encephalitis

Corresponding Author: Renaud DU PASQUIER

Version 0:

Reviewer comments:

Reviewer #1

(Remarks to the Author)

Perriot, Jones et al. present a timely study investigating the clonal expansion of neuron-reactive CD8+ T cells in the context of autoimmune encephalitis. The authors propose a novel methodological approach to detect neuron-reactive CD8+ T cells by using autologous hiPSC-derived neurons. They validate this methodology by demonstrating that CD8+ T cells can be activated by hiPSC-derived neurons in an antigen-dependent manner and confirm the presence of neuron-reactive CD8+ T cells in healthy donors.

Subsequently, using this method they identified neuron-reactive CD8+ T cell clonotypes in an anti-Ri AIE patient during two time points, either with acute disease activity or in remission. Furthermore, using scRNA and TCR sequencing, they confirmed the existence of this clonotype in both time points and report a specific gene signature for identification.

Additionally, a separate analysis using scRNA and TCR sequencing coupled with coculture experiments, identifies a neuron-reactive CD8+ T cell clonotype in a second anti-Ri AIE patient.

Overall, the study highlights the importance of CD8+ T cells in autoimmune encephalitis and provides novel insights the phenotype of neuron-reactive CD8+ T cells. The proposed method developed by the authors has the potential to advance the understanding of pathogenic CD8+ T cells in autoimmune encephalitis as well as other autoimmune diseases. However, some information is not adequately presented and some analyses could be performed more thoroughly.

- In addition to the detection clonal expansion of neuron-reactive CD8+ T cells in vitro, scRNAseq and TCRseq analysis was performed to further describe the phenotype of detected neuron-reactive clonotypes in two AIE patients. As it was sought to describe common features of identified neuron-reactive CD8+ T cell clonotypes, it is not clear why the analysis of both patients was performed separately. In order to compare the clonotypes of interest, the cluster composition and gene expression profile in both patients and find common features, samples should be integrated and analyzed together.
- In the scRNAseq and TCRseq analysis, CD8+ T cells were first clustered and subsequently, the most frequent clonotypes were determined and assigned to CD8+ T cell compartments. It is described that the neuron-reactive CD8+ T cell clonotype is the 8th most frequent clonotype among CD8+ T cells in the first patient and referred to Figures 4c and 4d (line 176ff.) However, both figure panels do not show the frequencies of clonotypes but the composition of clusters within the top 10 clonotypes. Please include information about the frequencies of the clonotypes in all samples analyzed.
- Additionally, scRNAseq and TCRseq analysis was performed on a sample from the first AIE patient at time of remission to compare the identified neuron-reactive CD8+ T cell clonotype in both timepoints. The transcriptional profile of the clonotype in both timepoints is compared and it is reported that the gene profile was similar with several genes being downregulated. To improve accessibility, the change in gene expression from active disease to remission should be displayed and assessed statistically in addition to Figures 5b and 5d.
- In part, the interpretation and discussion of results from the scRNAseq analyses should be revised as some conclusions are not supported by the presented results.
Line 277ff.: The identified neuron-reactive CD8+ T cell clonotypes are characterized by a high expression of genes like FAS or CXCR3. However, the expression of FAS among the clonotypes is not shown at all. Moreover, when looking Figure 4, about half of the cells from clonotype of interest (P1-C8) in the first patient are assigned to cluster 1 (Fig. 4c, d), which shows

only weak or no expression of CXCR3 (Fig. 4a, b).

Line 214ff., 278ff.: In the analysis of the second AIE patient, it is stated that the neuron-reactive clonotype (P2-C5) shows increased expression of KLRG1 and it is therefore concluded that the neuron-reactive CD8+ T cell gene signature includes increased expression of KLRG1. However, looking at Figures 6b and 6c, it is not evident that the KLRG1 expression is increased in this clonotype compared to other displayed clonotypes.

- The paper presents a novel and promising approach for detection of neuron-reactive CD8+ T cells with autologous hiPSC-derived neurons. However, it is not clear, why this method is only used for the first but not the second AIE patient although the abstract indicates it was applied to both patients. Additionally, analysis of the HLA allele potentially mediating antigen recognition by the CD8+ T cell clonotype identified in the scRNAseq analysis of the second AIE patient was not performed in an autologous system but solely with neurons derived from partially HLA-matched neurons, although the paper emphasizes the need to use an autologous system. If possible, the authors should repeat their experiment with the second AIE patient and include the data on clonality and entropy in their analyses. Otherwise, it should be stated in the paper why this experiment was not conducted and the abstract should be corrected in this aspect.

- In Figure 6d, the TCR of the CD8+ T cell clonotype identified in the scRNAseq analysis of the second AIE patient was tested for neuron-reactivity. By process of elimination, it is assumed that the antigen recognition by this TCR is mediated by an HLA-C allele. However, there is an inconsistency between the HLA-typing of the HDs displayed in Figure 6d and the corresponding Supplementary Table 3. In the Supplementary Table, HD6 is reported as positive for HLA-B*18:01 and HLA-B*44:03, while in the figure HD6 is reported as HLA-B*51:01-positive. If HD6 is positive for HLA-B*51:01, please correct the Supplementary Information. Otherwise, please correct Figure 6d. Additionally, if the latter is correct, it can therefore not be ruled out that antigen recognition might be mediated by HLA-B*51:01, which should be then considered in the conclusions and discussion or ruled out with further experiments.

- In histopathologic staining of brain tissue from the second AIE patient, the authors were able to detect CD8+ T cells expressing GZMB and TOX and therefore provided evidence, that expression of TOX is a feature of potentially pathogenic CD8+ T cells in AIE. As the scRNAseq analysis suggests that neuron-reactive CD8+ T cells are also characterized by KIR3DL1, additional staining for KIR3DL1 would strengthen the conclusions and provide a more comprehensive understanding of the pathogenic CD8+ T cells in AIE.

- The proposed method to detect neuron-reactive CD8+ T cells in an autologous system using hiPSC-derived neurons is first validated in HDs and then applied to one AIE patient in two timepoints. For this, the TCR-b repertoire ex vivo and after 14 days of coculture is analyzed for clonal expansion. In Figure 2c, TCR-b chains are depicted ex vivo and after coculture in HDs, with identical TCR-b chains colored consistently across both samples within a donor. Similarly, in Figure 3c, this consistency is intended for the samples from both timepoints of the first AIE patient. However, it is noted that while the same TCR-b chain is reported to be clonally expanded during both active disease and remission in this patient, the slices representing this chain are colored inconsistently. To enhance accessibility, it is recommended identical TCR-b chains are colored consistently within the patient.

- Please discuss implications and limitations of your study in more depth.

- Please add implications of your findings to the abstract.

- Please use consistent gene names in figures and text, e.g. IFGNRA in line 192 and IFGNR1 in Figures 5 and 6.

- Please provide the presented figures in a high-resolution quality.

- Gene names should be written in italic in figure legends.

- Please give references for the HLA allele frequencies (line 313ff.)

- Please provide information about data availability.

- Please provide information about statistical analyses performed in the study.

Reviewer #2

(Remarks to the Author)

In this study, the researchers developed a clever method for detecting neuron-reactive cytotoxic CD8+ T cells. They employed autologous iPSC-derived neurons treated with INF-gamma and TNF to induce the expression of HLA class I molecules. Initially, they convincingly demonstrated the neurons' ability to present peptides from infectious agents and to elicit strong CD8+ T cell responses. Subsequently, they cultured PBMCs from healthy donors with these autologous iPSC-derived neurons, successfully expanding neuron-reactive CD8+ T cells. More specifically, they used TCR-beta sequencing before and after culture to identify clones that have undergone significant expansion in contact with the neurons. The researchers then transferred these TCRs into Jurkat reporter cells to confirm their neuron-reactivity. They compellingly demonstrated that this reactivity is mediated through HLA class I, as blocking experiments effectively eliminated the reactivity.

In the next part of the study, they applied this methodology to identify neuron-reactive CD8+ T cells in a patient with anti-Ri autoimmune encephalitis. They successfully isolated a specific neuron-reactive clonotype and conduct single-cell RNA sequencing to obtain the gene expression profile of this clone (which includes TOX and GZMB). Additionally, they performed single-cell RNA sequencing on a second patient with anti-Ri encephalitis, demonstrating the expansion of CD8+ T cells that share the same phenotypic signature, and provide evidence that this TCR is also indeed neuron-reactive. In autopsy material from the same patient, they detected CD8+ T cells with this phenotype in the brain.

The methodology presented in the paper is highly original and holds significant relevance for the field of neuroimmunology with potential implications for broader areas of immunology (i.e. using iPSCs differentiated to the target cell of choice). The discovery of expanded CD8+ T cells with a cytotoxic gene-expression profile in two patients with anti-Ri encephalitis represents an important advancement in the under-researched domain of paraneoplastic autoimmune encephalitis. Their findings support the hypothesis that CD8+ T cells are key effector cells in autoimmune encephalitis with autoantibodies against intracellular epitopes. The methodology is sound and meet the expected standards, and there is enough detail provided in the methods for the work to be reproduced.

However, I have some concerns regarding the conclusions and claims in the study:

1. The authors claim that the CD8+ T cells are neuron specific (e.g. in the title, methods line 621, results line 158). However, they do not know the antigen, nor are they making any efforts to test to what extent the T cells/TCRs are indeed neuron specific. One way to do this would be to test the Jurkat T cells against non-neuronal cells, for instance against autologous iPSCs that have been differentiated into a non-neuronal cell type.

2. Although they use the healthy individuals to develop and test their assay, they do also use this group to compare the results to the two patients and make claims about it. For instance, from the abstract "This unbiased approach allowed for the identification of rare polyclonal neuron-reactive CD8+ T cells in healthy donors, and contrastingly, expanded clonotypes in two patients with anti-Ri AIE" or discussion "Both clonotypes were readily expanded ex vivo and belonged to the 10 most frequent clonotypes in each patient (Figures 4 and 6) contrasting with our findings in HD in whom neuron-reactive T cells were rare (frequency <0.01%, Figure 2)." My concern is that the healthy donors are on average half the age of the patients. It is well known that the presence of clonally expanded CD8+ T cells are much more common in aged individuals, and it has been shown that the presence of clonally expanded TOX+ CD8+ T cells are a common trait of "inflammaging" (<https://www.sciencedirect.com/science/article/pii/S1074761320304921>). Thus, the authors should consider including age-matched controls.

Regarding figures and clarity:

3. The font sizes and styles in the figures vary significantly, with some text appearing very small or in bold. I recommend that the authors standardize the font size and style to a readable, non-bold format for uniformity and clarity.

4. Results, line 116: "We considered a clonotype to be expanded in vitro if TCR- β chain frequency presented with a ≥ 9 -fold expansion as compared to ex vivo samples." What if the clonotype was not detected in the ex vivo samples? Was the frequency artificially set to a very low number? Or was only TCRs present in the ex vivo sample considered? This could be clarified in the methods.

Reviewer #3

(Remarks to the Author)

The manuscript by Perriot et al. reports about an innovative method to discover disease-specific neuronal-reactive CD8+ T cells. hiPSC-derived neurons have been used as APCs to look at the global autoreactive CD8+ T cell repertoire. Their unbiased approach was able to identify rare polyclonal neuron-reactive CD8+ T cells in healthy donors, and interestingly, expanded clonotypes in two patients with anti-Ri AIE. The ex vivo characterization of the clonotypes revealed a specific transcriptional program suggestive of a pathogenic potential of such cells.

General comments:

The manuscript is interesting and easy to follow although most of the figures (except for figure 1b) do not show any statistics.

According to the authors, the main discovery is the autoreactive clone, as deeply stressed in the title and in the discussion, nonetheless, they report findings from a screening of only 2 patients that might weaken such statement. On the other hand, the methodology developed is very interesting and novel.

The authors should strengthen their biological findings in terms of clone identification by including in the study more AIE patients or, alternatively, patients suffering from other autoimmune CNS-confined neurological disorders (e.g. multiple sclerosis, ADEM). Indeed, they should also prove and validate the importance of the identified clone through in vivo studies on animal models.

Detailed comments from the text:

The authors stated that: "Finally, we wondered whether these findings could be extended to other patients suffering from anti-Ri AIE. We thus profiled the CD8+ T cells from another patient (Ri02) with Ri-AIE by scRNAseq and conducted the

same analysis as for the previous patient. We identified 11 distinct clusters (Figure 6a and 6). Again, the 10 most frequent clonotypes were regrouped in the same clusters displaying an activated cytotoxic phenotype (i.e. KLRG1, PRF1, GZMB) (data not shown)".

It would be very helpful and would improve comprehension of the overall study to add a supplementary figure showing these data.

The authors stated that: "To assess whether P2-C5 was also neuron-reactive, we cloned this TCR into Jurkat cells and performed a coculture with partially HLA-matched neurons from HDs. Of the highest interest, we could confirm specific TCR activation against neurons expressing HLA-A*02:01 and HLA-C*14:02 (Figure 6d). Last, given the importance of TOX in triggering an encephalitogenic program in CD8+ T cells (24), we looked at whether we could find infiltrating TOX+ CD8+ T cells in the lesions of Ri02 on autopsy tissue. Strikingly, we could confirm the presence of infiltrating TOX+ CD8+ T cells thus suggesting the pathogenic role of the clonotype P2-C5. Additionally, CD8+ T cells present in the brain tissue were also GZMB+ and affixed to neurons (Figure 6f). This last set of experiments demonstrates that preponderant neuron-reactive CD8+ T cells with a cytotoxic encephalitogenic phenotype are most likely common findings in Ri-AIE and establish the role of CD8+ T cells in the pathogenesis of these disorders."

In figure 6E and 6F, the authors show only one representative picture. It would be greatly appreciated if they could include a proper quantification of the positive cells out of the total CD8 T cells.

The authors stated that: "Of note, Ri-AIE being a rare occurrence, we could only include two patients in our study including Ri02 from whom we had brain material, a unique asset. The fact that we identified highly activated neuron-reactive CD8+ T cells using two different approaches reinforces the validity of our findings for Ri-AIE overall. Indeed, our coculture assay with autologous neurons allowed us to identify the P1-C8 clonotype in Ri01. Assessment of the specific gene expression of this clonotype allowed us to identify a unique cytotoxic signature, which we were able to identify in a second patient with the same disease (Ri02). The success of identifying the P2-C5 clonotype in Ri02 using the signature from Ri01 demonstrates the strong association between this set of genes and pathology-associated CD8+ T cell clonotypes. There is thus a strong rationale to think that our findings may be extended to other patients with Ri-AIE and could provide a set of biomarkers to identify autoreactive CD8+ T cells in Ri-AIE. To our knowledge, this is the first demonstration in AIE of a specific CD8+ T cell phenotypic signature predictive of neuron-reactive TCRs."

I understand that this is a rare disease, and it is difficult to find additional patients, but, as I stated above, with only 2 patients the authors cannot claim to have a "strong rationale". I would, once more, encourage the authors to enlarge their cohort of patients including control groups with autoimmune CNS confined pathologies or try to get blood samples from international biobanks. Additional work on animal model of autoimmune CNS-confined disorders might also help to reinforce the overall claim of the manuscript.

Finally, figure 4 and the cloning identification through scRNAseq analysis is difficult to understand for the reader, maybe the authors should try to explain better this part of the study since it is of primary importance.

Version 1:

Reviewer comments:

Reviewer #1

(Remarks to the Author)

All of my comments have been addressed, the revision visibly improved the quality of the manuscript and the presented data.

Reviewer #2

(Remarks to the Author)

Pierrot et al. have addressed my previous concerns convincingly. However, their new comparison of CD8+ T-cell phenotypes in Ri-AIE versus AgD is based exclusively on single-cell statistics (Fig. 5). Because the hypothesis—and the paper's conclusions—concern differences between patient groups, I strongly recommend adding a patient-level differential-expression analysis (e.g., pseudobulk aggregation or a mixed-effects model with patient as a random effect). This will remove pseudoreplication and show whether the highlighted pathways are consistently altered across Ri-AIE patients.

Best regards,
Andreas Lossius

Reviewer #3

(Remarks to the Author)

It is difficult to determine whether the main objective of the paper is the description of the methodological approach or the actual comparison between healthy controls and patients with autoimmune encephalitis (AIE). Despite the scRNA-seq analysis of 47 ex vivo CD8+ T cells from a cohort of seven AIE patients and three age-matched controls, the comparisons shown are numerically very weak.

If the focus of the paper is the comparison between healthy and diseased individuals, the comparison between groups appears numerically underpowered.

In general, I would be more rigorous with the quantification of the immunofluorescence (IF) images, which currently seems a bit rough or imprecise.

On the other hand, the methodological protocol is clearly presented and follows a straightforward design, which facilitates reproducibility. While the methodological workflow appears robust, the comparative analyses are underpowered and limit the strength of the conclusions.

I believe the authors should consider conducting additional in vitro studies using neurons under various inflammatory conditions to better confirm their finding.

Specific points:

Fig1 B: in addition to the absolute MFI, the authors should calculate the MFI or the number of positive cells relative to DAPI (total number of cells). Indeed, at least by eye, the image of the post-stimulus condition show a higher number of cells.

Fig.1C : If it's a correlation, the interpolation line for the data is usually shown.

Why does the same virus elicit such a different response in different donors?

Fig. 3: the UMAP of neuron-specific CD8⁺ T cells presented in Fig. 3 includes data from only one patient and two controls, which represents a very limited sample size.

Fig. 4a: quantification is missing.

Fig. 4: conceptually, a single patient proves nothing. Since they recruited 7 patients, they should have included at least 3.

Fig. 6: quantification from a single sample doesn't make much sense. This should be removed. In the differential gene expression analysis comparing the transcriptomic profile between neuron-reactive KIR⁺ CD8⁺ Treg cells from AIE patients and AgDs, the cut off of $\log_{2}FC > 0.5$ is low, usually set at $\log_{2}FC > 1$.

Methods: the methods used for the scRNA-seq analysis are quite limited. For example, the statistical approaches and the tools used to quantify differences between cell populations under the two conditions are not described, and overall, the methodology is insufficiently explained.

Version 2:

Reviewer comments:

Reviewer #2

(Remarks to the Author)

Dear authors,

Thank you for carrying out the extra analyses. The new patient-level aggregation and mixed-effects modelling, along with the per-patient plots, fully resolve my concern about pseudoreplication.

To make these points fully transparent to readers, I have just two small wording requests:

1. Add one sentence to the Discussion noting that no genes were significant at the patient level in the pseudobulk/mixed-model analysis (3 controls vs 5 Ri-AIE), most likely because of limited sample size and uneven cell numbers.
2. Results, p. 7, lines 242–252: Where you write that several markers “displayed a significant increase/decrease,” please flag that the significance is at the single-cell level—for example, “(significant at the single-cell level)” after the first occurrence or add a brief footnote stating that significance in this section derives from single-cell tests.

These are purely textual clarifications; no further analyses are needed.

Best regards,
AL

Reviewer #3

(Remarks to the Author)

Although some of the questions raised before still remain in my view and in particular the low number of patients enrolled that might question the relevance of the study, the methodological approach developed is certainly of note and the authors have provided satisfactory answers, although not exhaustive, to the points raised in the previous round of revision.

Reviewer #1:

Perriot, Jones et al. present a timely study investigating the clonal expansion of neuron-reactive CD8+ T cells in the context of autoimmune encephalitis. The authors propose a novel methodological approach to detect neuron-reactive CD8+ T cells by using autologous hiPSC-derived neurons. They validate this methodology by demonstrating that CD8+ T cells can be activated by hiPSC-derived neurons in an antigen-dependent manner and confirm the presence of neuron-reactive CD8+ T cells in healthy donors.

Subsequently, using this method they identified neuron-reactive CD8+ T cell clonotypes in an anti-Ri AIE patient during two time points, either with acute disease activity or in remission. Furthermore, using scRNA and TCR sequencing, they confirmed the existence of this clonotype in both time points and report a specific gene signature for identification. Additionally, a separate analysis using scRNA and TCR sequencing coupled with coculture experiments, identifies a neuron-reactive CD8+ T cell clonotype in a second anti-Ri AIE patient.

Overall, the study highlights the importance of CD8+ T cells in autoimmune encephalitis and provides novel insights the phenotype of neuron-reactive CD8+ T cells. The proposed method developed by the authors has the potential to advance the understanding of pathogenic CD8+ T cells in autoimmune encephalitis as well as other autoimmune diseases. However, some information is not adequately presented and some analyses could be performed more thoroughly.

GENERAL ANSWER #R1: We thank Reviewer #1 for his/her thorough revision, positive feedback on our study, and the constructive comments. In response to his/her suggestions, we have undergone major revisions that have clearly improved the clarity and completeness of our analyses. In particular, we have increased the number of donors for the single cell RNA seq analysis from 2 to 10, including a total of 7 Ri patients and 3 sex- and age-matched controls. Thank to these additional experiments, we have been able to pinpoint key transcriptomic differences in neuron-reactive CD8+ T cells, revealing not only a distinct CD8+ T cells subset to be enriched in neuron-reactive cells, but also a dysregulated phenotype highly relevant to the context of Ri-AIE. Below, we provide a point-by-point response addressing each of the reviewer's comments and suggested revisions. Corrections in the manuscript appear underlined in yellow. Last, we edited the colors on figures to improve the reading, with everything related to controls in blue and everything related to disease in orange.

•#1 In addition to the detection clonal expansion of neuron-reactive CD8+ T cells in vitro, scRNAseq and TCRseq analysis was performed to further describe the phenotype of detected neuron-reactive clonotypes in two AIE patients. As it was sought to describe common features of identified neuron-reactive CD8+ T cell clonotypes, it is not clear why the analysis of both patients was performed separately. In order to compare the clonotypes of interest, the cluster composition and gene expression profile in both patients and find common features, samples should be integrated and analyzed together.

ANSWER #R1 comment #1: Initially, the study involved a separate analysis since we only identified a handful of markers identified in neuron-reactive CD8+ T cells. Having only 2 patients and no matched controls, we did not feel that we had the right experimental setup to go beyond this analysis. Yet, we acknowledge the importance of integrating all samples together to accurately

identify common features and gene expression profiles of neuron-reactive CD8+ T cell clonotypes. With the much larger number of numbers (in response to comments from Reviewers #2 and #3), we have now integrated and analyzed all samples together to better identify shared/distinct characteristics and compare gene expression profiles across groups. These expanded analyses confirm and further strengthen our conclusions regarding the dysregulated phenotype of neuron-reactive cytotoxic KIR+ CD8+ T cells in Ri AIE, particularly their loss of regulatory activity, which we hypothesize contributes to disease pathogenesis.

As a result of the added data, the second part of the results and of the discussion have been completely rewritten. Specifically, the following sections of the manuscript have been updated and expanded.

Results: from page 5, paragraph 2 to page 7 (lines 160-275), corresponding to Figures 4 and 5.

Supp. Tables 3 and 4 describe the cohort demographics. Supp table 5 describes the HLA matching across the entire cohort. Supp tables 6 and 7 includes the dysregulated genes identified in the different analysis.

Discussion: from page 7, paragraph 3 to page 10, paragraph 2 (lines 313-396)

Methods: p17, paragraph 1 “study population” (lines 539-543), p21, “single-cell RNA and VDJ sequencing” (lines 722-727; 734-737)

- #2 In the scRNAseq and TCRseq analysis, CD8+ T cells were first clustered and subsequently, the most frequent clonotypes were determined and assigned to CD8+ T cell compartments. It is described that the neuron-reactive CD8+ T cell clonotype is the 8th most frequent clonotype among CD8+ T cells in the first patient and referred to Figures 4c and 4d (line 176ff.) However, both figure panels do not show the frequencies of clonotypes but the composition of clusters within the top 10 clonotypes. Please include information about the frequencies of the clonotypes in all samples analyzed.

ANSWER #R1 comment #2: Following the additional experiments and the extended cohort, we have changed our approach regarding the analysis of the scRNA-seq data. Instead of comparing clonotypes within each donor (intra-subject approach), we have now moved on to a global comparative approach between three aged study subjects (AgD) and all 7 Ri AIE patients. Following Reviewer 1's comment, we have included information on *ex vivo* neuron-reactive clonotype frequencies between AgD vs Ri AIE, fitting with our current approach, rather than among the top 10 clones for each donor. We believe this approach offers a more comprehensive view of the clonotype frequencies across the entire cohort, as requested.

Corrections: Corresponding corrections are present in Supp. Figure 3b. We have performed an analysis presenting the TCR- β chain frequency as measured by bulk TCR- β chain repertoire sequencing of all clones from all neuron-reactive clones from each sample. The result section has been updated accordingly (lines 196-197)

- #3 Additionally, scRNAseq and TCRseq analysis was performed on a sample from the first AIE patient at time of remission to compare the identified neuron-reactive CD8+ T cell clonotype in both timepoints. The transcriptional profile of the clonotype in both timepoints is compared and

it is reported that the gene profile was similar with several genes being downregulated. To improve accessibility, the change in gene expression from active disease to remission should be displayed and assessed statistically in addition to Figures 5b and 5d.

ANSWER #R1 comment #3: In response to this comment, we have further detailed the comparison between two timepoints in the result section (lines 253-265) and Supp Figure 4. Specifically, we are mentioning the actual number of dysregulated genes, added a volcano plot displaying gene dysregulation as per Log2FC and adj. p value (Supp. Figure 4a), plotted the differential gene expression as a PCA to aggregate the information of the single cell level (Supp. Figure 4b), and as dot plot to compare individual gene expression between the two timepoints (the Log2FC from the total cohort analysis is shown as reference) (Supp. Figure 4c).

- #4 In part, the interpretation and discussion of results from the scRNAseq analyses should be revised as some conclusions are not supported by the presented results. Line 277ff.: The identified neuron-reactive CD8+ T cell clonotypes are characterized by a high expression of genes like FAS or CXCR3. However, the expression of FAS among the clonotypes is not shown at all. Moreover, when looking Figure 4, about half of the cells from clonotype of interest (P1-C8) in the first patient are assigned to cluster 1 (Fig. 4c, d), which shows only weak or no expression of CXCR3 (Fig. 4a, b). Line 214ff., 278ff.: In the analysis of the second AIE patient, it is stated that the neuron-reactive clonotype (P2-C5) shows increased expression of KLRG1 and it is therefore concluded that the neuron-reactive CD8+ T cell gene signature includes increased expression of KLRG1. However, looking at Figures 6b and 6c, it is not evident that the KLRG1 expression is increased in this clonotype compared to other displayed clonotypes.

ANSWER #R1 comment #4: We acknowledge that some statements were not properly supported by reported results on the figures, omitting some markers discussed in the text.

FAS, CXCR3 and KLRG1 are not discussed anymore in the manuscript following the entire reshuffling of the paper with the additional data. Yet, we have been careful to better match the results presented in the figures and discussed in the entire text. Additionally, we switched data presentation to a heatmap to able to cover more markers in the main figure.

In order to better characterize neuron-reactive clonotypes, we have thus performed thorough analyses comparing the cluster containing neuron-reactive clonotypes (new cluster 1 Figure 4) and other clusters. We added a supp table containing all dysregulated genes per cluster with log2FC and adj. p value for readers to consult (Supp table 6). The changes related to cluster characterization are found in the result section, lines 164-169. The data are presented in Supp figures 1 and 2. In addition, the in-depth characterization of the cluster containing the initially identified neuron-reactive clonotype is presented in the result section, lines 203-216, Figure 4g.

- #5 The paper presents a novel and promising approach for detection of neuron-reactive CD8+ T cells with autologous hiPSC-derived neurons. However, it is not clear, why this method is only used for the first but not the second AIE patient although the abstract indicates it was applied to both patients. Additionally, analysis of the HLA allele potentially mediating antigen recognition by the CD8+ T cell clonotype identified in the scRNAseq analysis of the second AIE patient was not performed in an autologous system but solely with neurons derived from partially HLA-matched neurons, although the paper emphasizes the need to use an autologous system. If possible, the

authors should repeat their experiment with the second AIE patient and include the data on clonality and entropy in their analyses. Otherwise, it should be stated in the paper why this experiment was not conducted and the abstract should be corrected in this aspect.

ANSWER #R1 comment #5: We were unable to perform an autologous neuron-PBMC co-culture for the second AIE patient (Ri02) due to a lack of sufficient PBMC. Indeed, generation of hiPSC to create neurons requires >2-3mio PBMC and we need >4-5mio PBMC to perform the subsequent co-culture. Since we only had 1 vial of 5mio PBMCs, we decided to prioritize phenotypical characterization of potential neuron-reactive clonotypes (based on the signature observed in Ri01). We therefore used these cells to isolate CD8+ T cells and perform subsequent scRNAseq analyses. Although we added several other donors to the study, autologous neuron-PBMC co-culture could still not be performed for the new donors because of the length of the procedure which would have taken another year and a half and thus not compatible with the time restriction associated with the manuscript's reviews. Nonetheless, this coculture technique has been central to identify all the neuron-reactive CD8+ T cells we found in subsequent experiments since we could identify this cluster enriched in neuron-reactive T cells using the data from Ri01 and the initial clonotype identified through neuron-PBMC coculture. The reviewer can rest assured that we firmly believe in the potential of this technique and currently have several studies based on leveraging this expertise. As a direct consequence for this manuscript and as suggested by Reviewer 1, we corrected the abstract to downplay the statements regarding autologous co-cultures (lines 43-55). The initial experiment setup also takes less space in the paper with the addition of the new scRNA seq data.

- #6 In Figure 6d, the TCR of the CD8+ T cell clonotype identified in the scRNAseq analysis of the second AIE patient was tested for neuron-reactivity. By process of elimination, it is assumed that the antigen recognition by this TCR is mediated by an HLA-C allele. However, there is an inconsistency between the HLA-typing of the HDs displayed in Figure 6d and the corresponding Supplementary Table 3. In the Supplementary Table, HD6 is reported as positive for HLA-B*18:01 and HLA-B*44:03, while in the figure HD6 is reported as HLA-B*51:01-positive. If HD6 is positive for HLA-B*51:01, please correct the Supplementary Information. Otherwise, please correct Figure 6d. Additionally, if the latter is correct, it can therefore not be ruled out that antigen recognition might be mediated by HLA-B*51:01, which should be then considered in the conclusions and discussion or ruled out with further experiments.

ANSWER #R1 comment #6: We thank reviewer #1 for highlighting this important misinterpretation from our side in Figure 6d. HD6 is indeed HLA-B*44:03+ and not HLA-B*51:01+ as described in the original figure 6d. We can therefore not rule out that HLA-B*51:01 may be involved in antigen presentation. Furthermore, subsequent analyses in other AgD and Ri patients using partially HLA-matched neurons did not allow us to restrain efficiently the possible HLA meditating antigen recognition. Since we were not in the position to confirm these data on the larger cohort, we have therefore removed any statements relative to the HLA-C restriction of these clones.

- #7 In histopathologic staining of brain tissue from the second AIE patient, the authors were able to detect CD8+ T cells expressing GZMB and TOX and therefore provided evidence, that expression of TOX is a feature of potentially pathogenic CD8+ T cells in AIE. As the scRNAseq analysis suggests that neuron-reactive CD8+ T cells are also characterized by KIR3DL1, additional staining

for KIR3DL1 would strengthen the conclusions and provide a more comprehensive understanding of the pathogenic CD8+ T cells in AIE.

ANSWER #R1 comment #7: We agree with reviewer #1 that the demonstration *in situ* (i.e. in the brain) that our signature (TOX **and** KIR3DL1) is present on CD8+ T cells would have been an asset for corroborating our findings in the periphery. While we initially performed immunohistochemistry staining for KIR3DL1 and found potential colocalization with CD8+ T cells, the results were inconclusive due to technical limitations, particularly due to an important signal emitted by astrocytes (see pictures below).

• #8 The proposed method to detect neuron-reactive CD8+ T cells in an autologous system using hiPSC-derived neurons is first validated in HDs and then applied to one AIE patient in two timepoints. For this, the TCR-b repertoire *ex vivo* and after 14 days of coculture is analyzed for clonal expansion. In Figure 2c, TCR-b chains are depicted *ex vivo* and after coculture in HDs, with identical TCR-b chains colored consistently across both samples within a donor. Similarly, in Figure 3c, this consistency is intended for the samples from both timepoints of the first AIE patient. However, it is noted that while the same TCR-b chain is reported to be clonally expanded during both active disease and remission in this patient, the slices representing this chain are colored inconsistently. To enhance accessibility, it is recommended identical TCR-b chains are colored consistently within the patient.

ANSWER #R1 comment #8: We agree with reviewer #1 that consistent coloring of identical TCR- β chains will undoubtedly improve reader comprehension regarding our data. This specific panel has been removed from the manuscript due to the extensive reshuffling. Yet, we have thoroughly checked all figures for congruency.

- #9 Please discuss implications and limitations of your study in more depth.
ANSWER #R1 comment #9: Newly generated data and subsequent analyses on additional 5 Ri-AIE patients and three aged matched controls have deeply strengthened our conclusions. Indeed, in this new version, we 1. confirm our initial findings that neuron-reactive CD8+ T cells from Ri patients express KIR3DL1 and TOX and; 2. further expand our conclusion demonstrating that these neuron-reactive CD8+ T cells are closely associated with regulatory KIR+ CD8+ T cells that exhibit impaired inhibitory function in Ri-AIE patients. Taking into account this large amount of new data, we have substantially modified our discussion which we believe to be much more exhaustive and improved. Furthermore, complying with Reviewer's #1 request, we have now included in the discussion sentences detailing our study limitations, in particular regarding the experimental approach we used in the study.

In text corrections: Discussion section. Lines 327-396

- #10 Please add implications of your findings to the abstract.

ANSWER #R1 comment #10: In line with our answer directly above, we have modified the abstract to be in accordance with our new findings. We have also briefly discussed the implications of our findings.

In text corrections: Lines 43-55

- #11 Please use consistent gene names in figures and text, e.g. IFGNRA in line 192 and IFGNR1 in Figures 5 and 6.

ANSWER #R1 comment #11: We have thoroughly reviewed our manuscript to ensure that gene names between text and figures are consistent.

- #12 Please provide the presented figures in a high-resolution quality.

ANSWER #R1 comment #12: We have now submitted high resolution figures.

- #13 Gene names should be written in italic in figure legends.

ANSWER #R1 comment #13: We have thoroughly reviewed our manuscript to ensure that gene names are written in italic in the figure, figure legends and text.

- #14 Please give references for the HLA allele frequencies (line 313ff.)

ANSWER #R1 comment #14: As we no longer support our statements relative to the sole HLA-C restriction of identified antigens, this sentence has been removed.

- #15 Please provide information about data availability.

ANSWER #R1 comment #15: A paragraph containing information about data availability is now included in the revised method section.

In text corrections: Lines 785-788

- #16 Please provide information about statistical analyses performed in the study.
ANSWER #R1 comment #16: A paragraph containing information about the statistical analyses is now included in the revised method section.

In text corrections: Lines 775-784

Reviewer #2:

In this study, the researchers developed a clever method for detecting neuron-reactive cytotoxic CD8+ T cells. They employed autologous iPSC-derived neurons treated with INF-gamma and TNF to induce the expression of HLA class I molecules. Initially, they convincingly demonstrated the neurons' ability to present peptides from infectious agents and to elicit strong CD8+ T cell responses. Subsequently, they cultured PBMCs from healthy donors with these autologous iPSC-derived neurons, successfully expanding neuron-reactive CD8+ T cells. More specifically, they used TCR-beta sequencing before and after culture to identify clones that have undergone significant expansion in contact with the neurons. The researchers then transferred these TCRs into Jurkat reporter cells to confirm their neuron-reactivity. They compellingly demonstrated that this reactivity is mediated through HLA class I, as blocking experiments effectively eliminated the reactivity.

In the next part of the study, they applied this methodology to identify neuron-reactive CD8+ T cells in a patient with anti-Ri autoimmune encephalitis. They successfully isolated a specific neuron-reactive clonotype and conduct single-cell RNA sequencing to obtain the gene expression profile of this clone (which includes TOX and GZMB). Additionally, they performed single-cell RNA sequencing on a second patient with anti-Ri encephalitis, demonstrating the expansion of CD8+ T cells that share the same phenotypic signature, and provide evidence that this TCR is also indeed neuron-reactive. In autopsy material from the same patient, they detected CD8+ T cells with this phenotype in the brain.

The methodology presented in the paper is highly original and holds significant relevance for the field of neuroimmunology with potential implications for broader areas of immunology (i.e. using iPSCs differentiated to the target cell of choice). The discovery of expanded CD8+ T cells with a cytotoxic gene-expression profile in two patients with anti-Ri encephalitis represents an important advancement in the under-researched domain of paraneoplastic autoimmune encephalitis. Their findings support the hypothesis that CD8+ T cells are key effector cells in autoimmune encephalitis with autoantibodies against intracellular epitopes. The methodology is sound and meet the expected standards, and there is enough detail provided in the methods for the work to be reproduced.

GENERAL ANSWER #R2: We are very grateful for the appreciative comments from Reviewer #2. Reviewer #2's comments regarding the importance of including age-matched controls (AgD) has led to important findings, which we believe substantially improve the relevance of our findings, as exposed in this revised manuscript. Corrections in the manuscript appear underlined in yellow. Last, we edited the colors on figures to improve the reading, with everything related to controls in blue and everything related to disease in orange.

However, I have some concerns regarding the conclusions and claims in the study:

1. The authors claim that the CD8+ T cells are neuron specific (e.g. in the title, methods line 621, results line 158). However, they do not know the antigen, nor are they making any efforts to test to what extent the T cells/TCRs are indeed neuron specific. One way to do this would be to test the Jurkat T cells against non-neuronal cells, for instance against autologous iPSCs that have been differentiated into a non-neuronal cell type.

ANSWER #R2 comment #1: We agree with Reviewer #2 that we cannot claim that our clonotypes are neuron-specific without testing these cells against non-neuronal cell types. In fact, we favored the term “neuron-reactive”, although some “neuron-specific” phrases have escaped our vigilance.

Following Reviewer #2’s comment, we therefore decided to test our neuron-reactive clonotypes against HLA-matched PBMCs rather than an hiPSC-derived non-neuronal cell types. Additionally, these cells are easier to culture, requiring less tedious steps than differentiating hiPSC into a non-neuronal cell type. Finally, as PBMC contain a diverse range of immune cells (eg. T lymphocytes, monocytes etc..), we believe that they may present a broader range of antigens. These experiments revealed that most of these clonotypes react also towards PBMC albeit at a significant lower magnitude. Importantly, 15% of these clonotypes did not recognize PBMCs thus making them more specific to neurons. Not being in the scope of this study, we did not investigate this further yet and kept the phrasing “neuron-reactive” clonotypes for all of them not to make overstatements regarding specificity.

Consequently, we have corrected all three “specific” phrasing to “reactive” (title, results section line 158 - now line 154, methods section, line 621 – now line 695).

2. Although they use the healthy individuals to develop and test their assay, they do also use this group to compare the results to the two patients and make claims about it. For instance, from the abstract “This unbiased approach allowed for the identification of rare polyclonal neuron-reactive CD8+ T cells in healthy donors, and contrastingly, expanded clonotypes in two patients with anti-Ri AIE” or discussion “Both clonotypes were readily expanded ex vivo and belonged to the 10 most frequent clonotypes in each patient (Figures 4 and 6) contrasting with our findings in HD in whom neuron-reactive T cells were rare (frequency <0.01%, Figure 2).” My concern is that the healthy donors are on average half the age of the patients. It is well known that the presence of clonally expanded CD8+ T cells are much more common in aged individuals, and it has been shown that the presence of clonally expanded TOX+ CD8+ T cells are a common trait of “inflammaging” (<https://www.sciencedirect.com/science/article/pii/S1074761320304921>). Thus, the authors should consider including age-matched controls.

ANSWER #R2 comment #2: We thank reviewer #2 for this very important point, which was instrumental in deepening our insights on our initial data. To address this issue, we have now included three age-matched donors (AgD) in our study in whom we performed scRNAseq of ex vivo CD8+ T cells.

These newly generated data enabled us to adequately compare the phenotype of CD8+ T cells between AgD and Ri AIE patients and demonstrate that:

1. Neuron-reactive CD8+ T cell clonotypes are common in aged donors, both controls and Ri-AIE patients. The cluster identifying cytotoxic KIR+ CD8+ regulatory T cells is specially enriched in neuron-reactive cells.
2. However, in Ri-AIE patients, there a significant decrease in *KIR* expression and the key regulatory transcription factor *IKZF2* (HELIOS), coupled with enhanced TCR signaling and increased cytokine (particularly *TNF* and *IFNG*) gene expression.

These newly generated data have led to significant changes in the figures and text which are detailed here below.

Results: from page 5, paragraph 2 to page 7 (lines 160-275), corresponding to Figures 4 and 5.

Supp. Tables 3 and 4 describe the cohort demographics. Supp table 5 describes the HLA matching across the entire cohort. Supp tables 6 and 7 includes the dysregulated genes identified in the different analysis.

Discussion: from page 7, paragraph 3 to page 10, paragraph 2 (lines 313-396)

Methods: p17, paragraph 1 “study population” (lines 539-543), p21, “single-cell RNA and VDJ sequencing” (lines 722-727; 734-737)

Regarding figures and clarity:

3. The font sizes and styles in the figures vary significantly, with some text appearing very small or in bold. I recommend that the authors standardize the font size and style to a readable, non-bold format for uniformity and clarity.

ANSWER #R2 comment #3: We have carefully standardized the formatting of the text throughout the manuscript, figures and supplementary materials.

4. Results, line 116: “We considered a clonotype to be expanded in vitro if TCR- β chain frequency presented with a ≥ 9 -fold expansion as compared to ex vivo samples.” What if the clonotype was not detected in the ex vivo samples? Was the frequency artificially set to a very low number? Or was only TCRs present in the ex vivo sample considered? This could be clarified in the methods.

ANSWER #R2 comment #4: This is an important technical aspect which was indeed not initially discussed. When the clonotype was not detected in ex vivo samples, we artificially set the ex vivo TCR- β chain frequency to a theoretical maximal frequency according to the number of CD8+ T cells that were sequenced. We have clarified this aspect in the methods section.

In-text corrections: Methods section, lines 659-661

Reviewer #3:

The manuscript by Perriot et al. reports about an innovative method to discover disease-specific neuronal-reactive CD8+ T cells. hiPSC-derived neurons have been used as APCs to look at the global autoreactive CD8+ T cell repertoire. Their unbiased approach was able to identify rare polyclonal neuron-reactive CD8+ T cells in healthy donors, and interestingly, expanded clonotypes in two patients with anti-Ri AIE. The ex vivo characterization of the clonotypes revealed a specific transcriptional program suggestive of a pathogenic potential of such cells.

General comments:

The manuscript is interesting and easy to follow although most of the figures (except for figure 1b) do not show any statistics.

According to the authors, the main discovery is the autoreactive clone, as deeply stressed in the title and in the discussion, nonetheless, they report findings from a screening of only 2 patients that might weaken such statement. On the other hand, the methodology developed is very interesting and novel.

The authors should strengthen their biological findings in terms of clone identification by including in the study more AIE patients or, alternatively, patients suffering from other autoimmune CNS-confined neurological disorders (e.g. multiple sclerosis, ADEM). Indeed, they should also prove and validate the importance of the identified clone through in vivo studies on animal models.

GENERAL ANSWER #R3: Following Reviewer's #3 important comment 3, and also complying with Reviewer #2' request comment 2 (aged controls), we have undertaken a massive enrollment, increasing the number of donors for whom we performed scRNA seq from 2 to 10 (5 additional Ri AIE patients and 3 sex/age-matched controls). Considering the rarity of the anti-Ri AIE, we respectfully consider that this represents an impressive improvement in the quality of our paper and we thank Reviewer #3 for having prompted us to make this additional effort.

In short, we confirm our initial findings that neuron-reactive CD8+ T cells from Ri-AIE patients express KIR3DL1 and TOX. But in addition, we are now able to demonstrate that these neuron-reactive CD8+ T cells are closely associated with regulatory KIR+ CD8+ T cells, which exhibit impaired inhibitory phenotype across Ri-AIE patients.

Corrections in the manuscript appear underlined in yellow. Last, we edited the colors on figures to improve the reading, with everything related to controls in blue and everything related to disease in orange.

Detailed comments from the text:

1. The authors stated that: "Finally, we wondered whether these findings could be extended to other patients suffering from anti-Ri AIE. We thus profiled the CD8+ T cells from another patient (Ri02) with Ri-AIE by scRNAseq and conducted the same analysis as for the previous patient. We identified 11 distinct clusters (Figure 6a and 6). Again, the 10 most frequent clonotypes were regrouped in the same clusters displaying an activated cytotoxic phenotype (i.e. KLRG1, PRF1, GZMB) (data not shown)".

It would be very helpful and would improve comprehension of the overall study to add a supplementary figure showing these data.

ANSWER #R3 comment #1: In Complying with comment#1 from Reviewer #3, as well as comment#4 from Reviewer #1 and comment #2 from Reviewer #2, we have reanalyzed the scRNAseq data from all donors, now including a total of 7 Ri-AIE patients (2 previously submitted + 5 new) and three age-matched controls (AgD) allowing for robust comparisons between neuron-reactive and non-reactive clonotypes across these donors. Therefore, instead of comparing clonotypes between themselves in each donor as initially performed, we have now moved on to a global comparative approach between AgD and Ri AIE patients and have removed figures related to “inter-clone” comparisons. In order to better characterize neuron-reactive clonotypes, we have performed thorough analyses comparing the cluster containing neuron-reactive clonotypes (new cluster 1 Figure 4) and other clusters. We added a supp table containing all dysregulated genes per cluster with log2FC and adj. p value for readers to consult (Supp table 6). The changes related to cluster characterization are found in the result section, lines 164-169. The data are presented in Supp figures 1 and 2. In addition, the in-depth characterization of the cluster containing the initially identified neuron-reactive clonotype is presented in the result section, lines 203-216, Figure 4g.

2. The authors stated that: "To assess whether P2-C5 was also neuron-reactive, we cloned this TCR into Jurkat cells and performed a coculture with partially HLA-matched neurons from HDs. Of the highest interest, we could confirm specific TCR activation against neurons expressing HLA-A*02:01 and HLA-C*14:02 (Figure 6d). Last, given the importance of TOX in triggering an encephalitogenic program in CD8+ T cells (24), we looked at whether we could find infiltrating TOX+ CD8+ T cells in the lesions of Ri02 on autopsy tissue. Strikingly, we could confirm the presence of infiltrating TOX+ CD8+ T cells thus suggesting the pathogenic role of the clonotype P2-C5. Additionally, CD8+ T cells present in the brain tissue were also GZMB+ and affixed to neurons (Figure 6f). This last set of experiments demonstrates that preponderant neuron-reactive CD8+ T cells with a cytotoxic encephalitogenic phenotype are most likely common findings in Ri-AIE and establish the role of CD8+ T cells in the pathogenesis of these disorders."

In figure 6E and 6F, the authors show only one representative picture. It would be greatly appreciated if they could include a proper quantification of the positive cells out of the total CD8 T cells.

ANSWER #R3 comment #2: We thank Reviewer #3 for this astute comment and have now quantified the proportion of infiltrating CD8+ T cells, GZMB+ CD8+ and TOX+CD8+ to provide a more comprehensive analysis. Additionally, we have included staining and quantifications from age-matched control brain tissues stained for CD8+ T cells, allowing for direct comparison with Ri-AIE patients. These new data further reinforce the potential pathogenic role of CD8+ T cells in Ri-AIE and enhance the robustness of our conclusions.

Figure corrections: Corresponding corrections are present in figure 6.

In-text corrections: Result section, lines 278-282

3. The authors stated that: "Of note, Ri-AIE being a rare occurrence, we could only include two

patients in our study including Ri02 from whom we had brain material, a unique asset. The fact that we identified highly activated neuron-reactive CD8+ T cells using two different approaches reinforces the validity of our findings for Ri-AIE overall. Indeed, our coculture assay with autologous neurons allowed us to identify the P1-C8 clonotype in Ri01. Assessment of the specific gene expression of this clonotype allowed us to identify a unique cytotoxic signature, which we were able to identify in a second patient with the same disease (Ri02). The success of identifying the P2-C5 clonotype in Ri02 using the signature from Ri01 demonstrates the strong association between this set of genes and pathology-associated CD8+ T cell clonotypes. There is thus a strong rationale to think that our findings may be extended to other patients with Ri-AIE and could provide a set of biomarkers to identify autoreactive CD8+ T cells in Ri-AIE. To our knowledge, this is the first demonstration in AIE of a specific CD8+ T cell phenotypic signature predictive of neuron-reactive TCRs."

I understand that this is a rare disease, and it is difficult to find additional patients, but, as I stated above, with only 2 patients the authors cannot claim to have a "strong rationale". I would, once more, encourage the authors to enlarge their cohort of patients including control groups with autoimmune CNS confined pathologies or try to get blood samples from international biobanks. Additional work on animal model of autoimmune CNS-confined disorders might also help to reinforce the overall claim of the manuscript.

ANSWER #R3 comment #3: As mentioned in our general response to Reviewer #3 (see above), we have expanded our cohort by including five additional Ri-AIE patients and three age-matched control study subjects, bringing the total to 7 Ri-AIE patients and 3 controls for the scRNA seq experiment. We were able to obtain PBMC from these five additional Ri-AIE patients thanks to collaborations with two internationally recognized expert centers: 1. Department of Neurology with Institute of Translational Neurology (University of Münster, Germany), Prof. Wiendl and 2. French Reference Center on Paraneoplastic Neurological Syndrome (Université Claude-Bernard, Lyon, France). Prof. Honnorat.

We have performed scRNAseq on *ex vivo* CD8+ T cells from all these samples and have reanalyzed the scRNAseq dataset with all samples integrated together. Additional TCR validation experiments enabled the identification of neuron-reactive CD8+ T cell clonotypes in all study subject, reinforcing our findings and ensuring reproducibility of our initial observations. Furthermore, addition of properly age-matched controls allowed us to adequately compare the two groups. The direct comparison highlighted key differences in neuron-reactive CD8+ T cells in Ri-AIE, which further strengthen the meaning and implications of our findings.

While, we fully agree with reviewer #3 that the demonstration of pathogenesis in an animal-based model (e.g. immunization with neuron-reactive CD8+ T cells in HLA-humanized mice) would provide compelling evidence regarding the role of these cells, we respectfully think that including animal-based experiments would have gone beyond the scope of this already very consistent paper.

Including these donors led to a major rewriting of the manuscript and presentation of major novel data. As a result of the added data, the second part of the results and of the discussion have been completely rewritten. Specifically, the following sections of the manuscript have been updated and expanded.

Results: from page 5, paragraph 2 to page 7 (lines 160-275), corresponding to Figures 4 and 5.

Supp. Tables 3 and 4 describe the cohort demographics. Supp table 5 describes the HLA matching across the entire cohort. Supp tables 6 and 7 includes the dysregulated genes identified in the different analysis.

Discussion: from page 7, paragraph 3 to page 10, paragraph 2 (lines 313-396)

Methods: p17, paragraph 1 “study population” (lines 539-543), p21, “single-cell RNA and VDJ sequencing” (lines 722-727; 734-737)

4. Finally, figure 4 and the cloning identification through scRNAseq analysis is difficult to understand for the reader, maybe the authors should try to explain better this part of the study since it is of primary importance.

ANSWER #R3 comment #4: We acknowledge that the transition between the phenotypical description of the neuron-reactive clone in figure 4 and ulterior scRNAseq analysis in figure 5 may lack some clarity. In our revised manuscript, we have attempted to ensure more congruence between the figures and provide enough details in the text and figure legends for the reader to follow more easily. Also, the methodology to select clonotypes is much more straightforward. Yet, we specifically added a scheme to explicit better this rationale (Figure 4a) and detailed the selection process (result section lines 174-191)

RESPONSE TO REVIEWER COMMENTS

Reviewer #1 (Remarks to the Author):

All of my comments have been addressed, the revision visibly improved the quality of the manuscript and the presented data.

We thank Reviewer 1 for their initial comments and are pleased to see that they are satisfied with the revisions we have performed.

Reviewer #2 (Remarks to the Author):

Pierrot et al. have addressed my previous concerns convincingly. However, their new comparison of CD8⁺ T-cell phenotypes in Ri-AIE versus AgD is based exclusively on single-cell statistics (Fig. 5). Because the hypothesis—and the paper's conclusions—concern differences between patient groups, I strongly recommend adding a patient-level differential-expression analysis (e.g., pseudobulk aggregation or a mixed-effects model with patient as a random effect). This will remove pseudoreplication and show whether the highlighted pathways are consistently altered across Ri-AIE patients.

Best regards,

Andreas Lossius

ANSWER: We thank Reviewer 2 for this additional comment to ensure that our findings are consistent across donors and that the alterations observed are not due to pseudoreplication. We have now performed a pseudobulk differential expression analysis, as well as a mixed-effects model run on the cell-level with donor as a random-effect using the MAST software. Yet, we do not find any notable significant differential expression in either case.

Despite the necessity to control for pseudoreplication, aggregated at the donor level, our analysis in figure 5 has just sufficient samples for bulk RNA-seq differential expression analyses, with 3 control samples versus 5 Ri samples. Additionally, there are vastly different numbers of cells per donor (a minimum of 26 and maximum of 395 cells), leading to a greater than 10-fold difference in the total number of reads per pseudo-bulked sample. We still see many genes with zero counts in the pseudo-bulked smaller samples, which is one of the statistical issues with single-cell gene counts that is mitigated by aggregating counts across sufficient numbers of cells. Considering the heterogeneity expected between Ri-AIE patients and even between the age-matched donors (AgD), the small sample size and uneven cell numbers, it is unsurprising that we did not find significant differences in the pseudobulk analysis. Thus, we believe that the absence of significance observed in the pseudobulk and MAST analysis is more due to power limitations of the dataset rather than to an absence of biological differences.

So, to ensure that the highlighted genes were consistently altered across patients, we generated graphs showing the gene expression for each gene individually at the donor level using pseudobulked aggregated data and measured as fragments per million (counts corrected for differences in fragment number between donors). This data representation shows that most of the genes highlighted in the paper are consistently altered in Ri-AIE, despite some interindividual variability (see supplementary figures 4 and 5). Of note, one or two patients (not always the same) are sometimes seen behaving differently but the majority

of the Ri-AIE patients show the same trends as the single-cell statistical analysis. In consequence, we believe that the statistically significant differences observed at the single-cell level remain biologically relevant despite the limitations of the analysis at the donor level. To ensure transparency, we have included these graphs as supplementary figures and tempered our conclusions.

Furthermore, since some genes displayed discrepant trends between the single-cell and the donor level analyses, we decided to remove them from the Figure 5 and not to consider them altered in Ri-AIE patients to ensure robustness of our analyses and conclusions. The genes removed are: *KLRC1* (Fig. 5c), *EGR3* (Fig. 5d), *CBL*, *DOK2* and *DGKE* (Fig. 5e), *HLA-DRB5* and *VCAM1* (Fig. 5f), *GZMA* and *FASLG* (Fig. 5g). We also revised the corresponding Supplementary Figure 6 in the current revised manuscript.

Last, the analysis presented in Figure 5 previously included all cells from the neuron-reactive clonotypes, yet further strengthen our conclusions and as a matter of congruence with the rest of the manuscript, we now focused this analysis on the cells from cluster 1 only for these clonotypes. We have thus updated the Figure 5 accordingly. It changes very little in the analysis, with pathways mainly remaining the same (only leishmaniasis replaces EBV infection) and we lost significance for 3 genes (*CD44*, *NKG7* and *CSF1*).

Following these analyses, we have made the following modifications in the manuscript:

Abstract, p2, line 49: “Intriguingly, KIR+ CD8+ T cells from most Ri-AIE patients presented with a significant decrease”

Results, p6, lines 227-229: “We thus performed a differential gene expression analysis comparing the transcriptomic profile between neuron-reactive KIR+ CD8+ Treg cells in cluster 1 from Ri-AIE patients and AgDs. We found 238 upregulated genes ($\log_{2}FC > 0.5$, adj-p value < 0.05) and 437 downregulated genes ($\log_{2}FC < -0.5$, adj-p value < 0.05; Figure 5a, Supplementary Table 7).”

Results p.6, lines 230-231: “The top 5 most enriched KEGG pathways corresponded to antigen processing and presentation, Leishmaniasis, Epstein-Barr virus infection, rheumatoid arthritis, Th17 cell differentiation, and human T-cell leukemia virus 1 infection (Figure 5b).

Results, p7, lines 242-252: “Consistent with this decrease of inhibitory receptors, neuron-reactive KIR+ CD8+ T cells from Ri-AIE displayed a strong increase in genes associated with TCR activation (*FOS*, *FOSB*, *JUN*, *JUND*, *RELB*, *EGR1*, *EGR2*, *EGR3*; Figure 5d) and a global trend to a decrease in genes active in TCR signaling inhibition (*INPP5D* (*SHIP1*), *CBLB* being significantly downregulated; Figure 5e). We next investigated if this observation translated into an increase in activation markers and effector function. As expected, most of the activation markers studied displayed a significant increase as compared to neuron-reactive KIR+ CD8+ T cells from AgDs (*HLA-DQA1*, *HLADQB1*, *HLA-DRA*, *HLA-DRB1*, *HLA-DRB5*, *CD44*, *CD69*, *CD74*, *ICAM1*, *VCAM1*; Figure 5f). Interestingly, we observed a significant increase in genes involved in pro-inflammatory cytokine and chemokine production (*IFNG*, *TNF*, *CCL3*, *CCL4*, *CSF1*) together with a decrease in some granzymes genes (*GZMA*, *GZMH*) while other granzymes were not statistically downregulated (*GZMB*, *GZMK*). Additionally, these cells displayed a decrease in *GZMH* while they still retained an important expression of other cytotoxic genes (*GZMK*, *FASLG*, *LAMP1*, *NKG7*) (Figure 5g).”

Results, p7, lines 255-258: “In order to assess interindividual variability, we aggregated all cells from each donor to conduct a pseudobulk analysis at the donor level. This analysis showed consistent patterns across multiple donors and evidenced trends confirming the analysis performed at the single cell level (Supplementary Figures 4 and 5).”

Discussion, p10, lines 396-398: “Of note, we observed some degree of variability among Ri-AIE patients with alterations in the aforementioned pathways being more marked in some patients than others thus warranting further confirmation in a larger cohort of patients.”

Methods, p22 lines 806-812: “For sample-level analyses, we used Seurat’s Aggregate Expression to calculate pseudobulk counts and DESeq2 (doi:10.1186/s13059-014-0550-8.) for testing for differential expression. In both cases, tests were run via Seurat’s FindMarkers function, with an average log-fold change cutoff of 0.5 and a Benjamini-Hochberg false discovery rate of 0.05 used as significance cutoffs in cell-level analyses. For dot plots, we used DESeq2’s fpm function directly to calculate fragments per million for pseudobulk counts, to normalize for differences in fragment counts between samples.”

Addition of two supplementary figures:

Supplementary Figures 4 : Expression of genes associated with KIR Treg, TCR activation and TCR inhibition per donor (pseudobulk aggregation).

Supplementary Figure 5 : Expression of genes associated with T cell activation, cytokine production/cytotoxicity and TOX pathway per donor (pseudobulk aggregation).

We also updated the Supplementary table 7 to match the current data.

Reviewer #3 (Remarks to the Author):

We thank Reviewer 3 for taking the time to provide their feedback on our revised manuscript. We have carefully considered each of the comments and have made every effort to address them comprehensively. In some cases, the references to figures/data or specific remarks were unclear, but we have tried to address their comments to our best. We hope that they will answer the Reviewer’s concerns.

Comment 1

It is difficult to determine whether the main objective of the paper is the description of the methodological approach or the actual comparison between healthy controls and patients with autoimmune encephalitis (AIE).

The aim of this study is indeed two-sided: First, to present a novel methodology for the identification and characterization on neuron-reactive CD8+ T cells and second, to demonstrate its usefulness in a pathologically-relevant context, here anti-Ri AIE.

To take into Reviewer #3 comment, we have now clarified these objectives in the introduction section page 3, Lines 97-99: “Our primary objectives were to first establish and validate this novel method, and second to leverage this approach for assessing the phenotype of neuron-reactive CD8+ T cells in the context of Ri-AIE.”

Comment 2

Despite the scRNA-seq analysis of 47 *ex vivo* CD8⁺ T cells from a cohort of seven AIE patients and three age-matched controls, the comparisons shown are numerically very weak.

If the focus of the paper is the comparison between healthy and diseased individuals, the comparison between groups appears numerically underpowered.

We acknowledge Reviewer 3's comment, however, we are not quite sure to understand whether the Reviewer refers to the number of donors enrolled or to the number of cells analyzed. In addition, we were not able to pinpoint what this number "47" is referring to.

Nevertheless, we did our best to comprehensively answer Reviewer 3's comment.

If Reviewer 3 is referring to the number of patients, we would like to highlight that it is precisely upon Reviewer 3's prior suggestion (as part of the first revision round), that we substantially expanded our cohort from 2 to 7 anti-Ri AIE patients. Indeed, although the study was initially meant as a proof-of-concept study (fully in line with Nature Communications' scope), Reviewer 3 specifically asked to increase the size of our cohort. We found this comment sound and agreed with the Reviewer that it would indeed dramatically improve the strength of our study. Therefore, we deployed tremendous efforts to find samples from additional patients and then perform a series of new experiments and analyses, which took a whole year to complete. Given the rarity of this disease, it is only thanks to international collaborations with the University Hospitals from Lyon and Munster that we were able to achieve such a challenge. Again, given the rareness of Ri-AIE, (e.g. the whole French Ri-AIE cohort consisted in only 36 patients in 2020 (1), and given the fact that we had to obtain viable PBMC in a sufficient number for the planned experiments, we respectfully believe that we fully addressed Reviewer 3's requests as emitted in the first round of revision.

If, however, Reviewer #3 is referring to the number of cells analyzed, then we would like to point out that our study is not based on 47 *ex vivo* CD8⁺ T cells. We have analyzed 48 (and not 47) clonotypes (27 from controls and 21 from Ri-AIE patients), which represent a total of 3'789 individual cells (i.e. data points), 2'317 cells from control and 1'472 cells from Ri-AIE patients. The reviewer may have been misled by Figure 4d, which displays the proportion of clonotypes tested (3.9% of all clonotypes from cluster 1 in average). In fact, these 48 clonotypes represent 49.1% of AgD expanded T cells from cluster 1 and 23.7% of Ri-AIE expanded CD8⁺ T cells from cluster 1. We have changed the percentage in the manuscript to be expressed as percentage of expanded CD8⁺ T cells and not total CD8⁺ T cells. Thus, these clonotypes are representative of the cells within this cluster.

To further improve the clarity of our manuscript and to avoid any confusion, we have proceeded to the following changes:

Results, p5, lines 183-184: "We found 488 individual clonotypes from 5'014 cells in all AgDs (average of 162.7 clonotypes/donor) versus 696 clonotypes from 6'584 cells in Ri-AIE (average of 99.4 clonotypes/donor, Figure 4d)."

Results, p5, lines 192-193: "We selected a total of 48 clonotypes, 27 from AgDs (49.1% of all expanded CD8⁺ T cells *ex vivo* in cluster 1 from AgDs) and 21 from Ri-AIE (23.7% of all expanded CD8⁺ T cells *ex vivo* in cluster 1 from Ri-AIE),"

We edited Figure 4d to include pie charts with the cell numbers, updated the figure legend in consequence: p14, line 483“d) Pie charts representing the distribution of clonotypes (left) and cells (right) from AgD subjects (top) and Ri-AIE patients (bottom) present in cluster 1.”

Comment 3

In general, I would be more rigorous with the quantification of the immunofluorescence (IF) images, which currently seems a bit rough or imprecise.

In our manuscript, the only IF image-based quantification relates to Figure 6. Yet, this quantification has been performed rigorously, counting the numbers of cells per area in a precise manner. If the reviewer refers to Figure 1b, the quantification was done by flow cytometry as already clearly stated in the figure legend (see our response to the specific points, first comment from the Reviewer).

Comment 4

On the other hand, the methodological protocol is clearly presented and follows a straightforward design, which facilitates reproducibility. While the methodological workflow appears robust, the comparative analyses are underpowered and limit the strength of the conclusions.

We thank the Reviewer for acknowledging the clarity and robustness of our methodological workflow. We are aware that the number of subjects in our study is not as high as it could be for more prevalent diseases, such as multiple sclerosis. Obviously, having more study patients and control subjects may have further strengthened our findings. However, we believe that our findings on the nature of neuron-specific CD8+ T cells in Ri-AIE patients not only illustrate the new methods described here, but are also directly relevant for the understanding of the immunopathogenesis of Ri-AIE. We also would like to emphasize that AIE are rare neurological conditions with an estimated incidence of 3.6 to 8.9 per million person-year (2), and Ri-AIE is just a subgroup of this population, not even the most prevalent one. Of note, several recent studies published in Nature Communications and Nature Medicine that focused on rare diseases (for e.g. some rare forms of malignancies) reported similar group sizes (typically 3-5 cases per group) (3-5). We believe that our study aligns favorably with those and provides very contributive data on the immunopathogenesis of Ri-AIE within the constraints of disease rarity. Undoubtedly, more studies will be necessary to confirm to which extent our findings apply to the larger population of Ri-AIE or AIE patients but it does not seem correct to refute them here on the basis of the cohort size.

Nevertheless, to mitigate our assertions, we have now modified the revised manuscript, as following:

Discussion, p10, line 390 “Taken together, our findings suggest that in AIE, autoreactive KIR+ CD8+ T cells most likely shift from their original purpose of containing autoimmunity towards infiltration of the brain, thus contributing to the pathogenesis of neurological autoimmune disorders.

Discussion, p10, lines 396-398: “Of note, we observed some degree of variability among Ri-AIE patients with alterations in the aforementioned pathways being more marked in some patients than others thus warranting further confirmation in a larger cohort of patients.”

Comment 5

I believe the authors should consider conducting additional in vitro studies using neurons under various inflammatory conditions to better confirm their finding.

In order to fully understand the pathogenesis of Ri-AIE, we agree that functional studies will be needed at some point to explore how these CD8+ T cells interact with neurons and damage them. However, functional studies are out of scope for this precise study aiming at finding and characterizing neuron-reactive CD8+ T cells. Our objective was already quite ambitious given the huge workload it entailed. Indeed, among other things, we had to reprogram erythroblasts into hiPSCs, differentiate them into neurons, perform coculture experiments, reconstruct TCR of neuron-reactive CD8+ T cells, validate them, perform RNA single cell sequencing on selected clonotypes and conduct complex bio-informatic analyses. Furthermore, and most importantly, not only are Ri-AIE patients rare, but PBMCs that we could obtain from different collaborations were available in a very limited number, thus drastically limiting the extent of possible experiments.

To acknowledge this limitation and delineate potential future directions, we have introduced the following sentence in the

Discussion, p10, lines 392-394: “Functional studies would be warranted to further explore how these autoreactive KIR+ CD8+ T cells directly cause the damages seen in Ri-AIE lesions.”

Specific points:

Comment 6

Fig1 B: in addition to the absolute MFI, the authors should calculate the MFI or the number of positive cells relative to DAPI (total number of cells). Indeed, at least by eye, the image of the post-stimulus condition show a higher number of cells.

Fig 1b is related to the MFI measured by flow cytometry and is not based on a microscopy analysis. Given that flow cytometry allows the acquisition of data “per cell” already, no additional DAPI counterstaining is required in this case. Our analysis is sound and follows state-of-the-art flow cytometry practices. The method for this analysis is already explained in the Methods and in the figure legend:

Figure legend: “b) Flow cytometry analysis of HLA class I expression (anti-HLA-ABC antibody, (clone W6/32)) of hiPSC-derived neurons from 6 healthy donors (HD1-HD6) untreated (control) or treated with IFN γ and TNF α for 48 hours (Wilcoxon test, p value = 0,0156).”

Methods section: “Flow cytometry. For HLA class I assessment, neurons were detached using TrypLE (Gibco) and washed with phosphate buffer saline (PBS) and FBS (2:100). Cells were then stained with a pan HLA-A, -B, -C antibody (2:100, Santa Cruz Biotechnology, clone W6/32), washed again with PBS + 2% FBS and fixated in PBS + PFA (4:100, Electron Microscopy Sciences).[...] Surface marker expression was assessed using an LSR II flow cytometer (BD Biosciences).”

As per Figure 1a, it is indeed an IF staining. But here, the goal of the figure 1a is “only” to illustrate visually the massive increase of HLA class I expression on neurons in inflammatory conditions (IFN-g and TNF-a), this is thus intended as a visual illustration (the same is true for Figure 3a). By contrast, Figure 1b, which reflects flow cytometry acquisition of data, allows for statistical analyses, as mentioned above.

Comment 7

Fig.1C : If it's a correlation, the interpolation line for the data is usually shown.

We thank Reviewer 3 for this remark and have added the line on the graph in the revised **Figure 1c.**

Comment 8

Why does the same virus elicit such a different response in different donors?

The immunological response against viral epitopes is highly personal and depends on many parameters, including previous exposure to the virus and HLA alleles expressed by an individual for example. In this assay, we used a well characterized pool of peptides selected to be immunodominant for given HLAs (the list of epitopes with presenting HLAs and donor matching is detailed in Supplementary Tables 1 and 2). Thus, the observed variability in CD8+T cell response across donors reflects the fact that the HLA class I molecules are different from an individual to the other, as determined genetically, and thus can induce a different magnitude of response in front of a given virus. Furthermore, the prior exposure to a given virus varies between individuals and thus the adaptive cellular immune response varies accordingly, such as shown in several labs including ours (6, 7).

Comment 9

Fig. 3: the UMAP of neuron-specific CD8⁺ T cells presented in Fig. 3 includes data from only one patient and two controls, which represents a very limited sample size.

We do not present any UMAP in Figure 3. Instead, Figure 3 illustrates the results of our autologous co-culture experiment using PBMCs and hiPSC-derived neurons from a single Ri-AIE patient (Ri01).

If the reviewer refers to Figure 4, we can assure the reviewer that data from all 7 patients and 3 age-matched controls are displayed.

This is specified in the figure legend “*b) UMAP displaying unsupervised clustering of ex vivo CD8+ T cells from three aged donors (AgD) and seven Ri-AIE patients assessed by scRNAseq resulting in 16 different clusters*”.

If the reviewer is confused to the coloring in Figure 4c, this representation serves only to highlight the cluster 1 (green), and the neuron-reactive clonotype previously identified in Figure 3, now contextualized within the larger cohort. In Figure 4c, this clonotype appears in red), among all CD8+ T cells (grey). In any case, all data points from this UMAP are the result of pooling of all 10 donors assessed (7 Ri AIE and 3 AgD).

This is already specified in the figure legend 4c “*c) Highlighting of cluster 1 (green) and CD8+ T cells presenting with the same TCR-β chain CDR3 amino acid sequence as the strongly expanded neuron-reactive clonotype from Ri01 (red)*.”

Comment 10

Fig. 4a: quantification is missing.

Figure 4a is a scheme, here, the only quantification we had to perform is the number of donors in each group, which is displayed (n=3 AgD; n=7 Ri-AIE).

If the reviewer refers to Figure 3a (images of neurons stained for HLA), we did not include quantification as we believe that the fluorescence intensities present on the picture are sufficient to set our case. In addition, we cannot perform statistical analysis on 1 patient. Nevertheless, to comply with what may be the request of Reviewer 3, we have now added the fold change in MFI related to the HLA expression in between the 2 conditions as quantified by flow cytometry (same methodology as for Figure 1b).

Results, p5, lines 150-151: “As described for HDs, we generated hiPSC-derived neurons that upregulated HLA class I molecules upon IFN γ and TNF α exposure (43.4 fold change as measured by flow cytometry(data not shown), Figure 3a for fluorescence microscopy observations).”

Comment 11

Fig. 4: conceptually, a single patient proves nothing. Since they recruited 7 patients, they should have included at least 3.

Figure 4 displays the data acquired for the entire cohort, i.e. 3 AgD controls and 7 Ri-AIE patients. Therefore, we assume that the Reviewer refers to Figure 3 in their comment. Additionally, we are unsure if Reviewer 3 is referring to the biological validity of our findings regarding the frequency/phenotype of neuron-reactive CD8 $^+$ T cells; or to the technical validity of our method to identify neuron-reactive CD8 T cells through coculture of PBMCs with autologous hiPSC-derived neurons.

If the reviewer refers to the biological findings, the demonstration on one patient here was indeed an exploratory proof-of-concept. We agree that a single patient is not sufficient to prove our point and we do not build any conclusions in the manuscript based on this patient alone, but rather on the entire cohort. The data acquired in this patient (Figure 3) served as a basis for establishing selected experiments in 6 additional Ri-AIE patients and 3 AgD, experiments that confirmed our initial findings in this first patient, as illustrated in Figure 4. Last, although we added several other donors to the study, autologous neuron-PBMC co-cultures could not be performed for these new donors due to the length of the procedure which would have taken another year or more. Furthermore, as mentioned earlier, we would anyway not have had enough PBMCs to perform these co-cultures.

If, however, the Reviewer 3's comment relates to the outcome of the coculture methodology to identify neuron-reactive CD8 T cells, we would like to emphasize that our claims and conclusions do not stand on one patient only but on the findings from the same experiment performed on 6 additional healthy controls (Figure 2), which assert the technical robustness and reproducibility of our strategy.

Comment 12

Fig. 6: quantification from a single sample doesn't make much sense. This should be removed.

In the first round of revision, Reviewer 3 made the following comment:

“In figure 6E and 6F, the authors show only one representative picture. It would be greatly appreciated if they could include a proper quantification of the positive cells out of the total CD8 T cells.”

Reviewer 3 asked in the first round of revision to provide more data on Figure 6, but not to remove it. Based on Reviewer #3's initial comments, we have performed accurate quantification for the entire figure and have included controls when possible (total CD8+ T cell quantification).

Overall, we agree that this is only one patient. But for all the reasons pertaining to the rarity of this disease (rarity of Ri-AIE, scarcity of brain biopsies in this indication, difficulty to obtain good quality brain samples at autopsy, etc), doesn't Reviewer 3 agree that having been able to obtain brain samples from one Ri-AIE patient is already an asset and a valued contribution to this manuscript? Although it is only one patient, we still believe that these data are in concordance with the other parameters assessed in this study, as well as in concordance with the known neuropathological features in Ri-AIE (8). The identification of TOX expression in CD8+ T cells within brain lesion of an Ri-AIE patient provides new information that may help generate future hypotheses in the field. Thus we consider that this figure 6 should remain in the manuscript as it offers to the scientific community additional piece of evidence reinforcing the hypothesis on the role of specific CD8+ T cell subsets in the pathogenesis of Ri-AIE.

Comment 14

In the differential gene expression analysis comparing the transcriptomic profile between neuron-reactive KIR+ CD8+ Treg cells from AIE patients and AgDs, the cut off of $\log_{2}FC > 0.5$ is low, usually set at $\log_{2}FC > 1$.

We thank Reviewer 3 for this comment. Cut off at 0.5 or 1 are both frequently used in studies presenting single cell RNA seq experiments. Thus our study is in line with current practices in the field. Here, we used the $\log_{2}FC > 0.5$ cut off only for the pathway analysis.

To comply with Reviewer 3 remark, we performed the same analysis using $\log_{2}FC > 1$ as cut off (selecting thus 203 genes instead of 238 with the 0.5 cut off). We looked at the top 10 pathways identified. Eight out of 10 pathways were found to be the same (4 out of 5 for the top 5). When looking at the genes involved in these pathways and identified in our data set, to assess redundancy, we found more than 86% of homology between the analysis with a cut off of 0.5 vs 1. More importantly, of all the genes specifically presented in the heatmaps in Figure 5, only CD74 was absent in the analysis done with a cut off of 1 instead of 0.5. In conclusion, using a $\log_{2}FC > 0.5$ or $\log_{2}FC > 1$ give very similar results with only a minor change which does not change the interpretation of these results.

Comment 15

Methods: the methods used for the scRNA-seq analysis are quite limited. For example, the statistical approaches and the tools used to quantify differences between cell populations under the two conditions are not described, and overall, the methodology is insufficiently explained.

We thank Reviewer 3 for this comment. We have thus substantially expanded the method section to provide more details, such as:

Methods, p21 lines 755-766: “CD8+ T cell clustering analysis and cell subtype annotation

Ambient RNA contamination was removed from the Cell Ranger counts using Cellbender v0.3.0 (<https://doi.org/10.1038/s41592-023-01943-7>), with the number of cells detected per sample from the initial Cell Ranger analysis used as the “expected-cells” parameter and 25000 as the “total droplets included” parameter. TCR-sequences were collated with cell barcodes using scRepertoire v2.3.2, quantifying all productive TCR chains. Filtered counts generated by Cellbender were further filtered to select cells with a single TCR- β chain sequence in the VDJ reads, maximum 10% mitochondrial reads and at least 100 genes expressed; and genes expressed in at least 10 cells; 109’478 cells (4’824 – 13’569 per donor) remained after quality filtering. These cells were integrated and clustered using Seurat’s atomic sketch integration (54) with 5000 cells used for sketching, 20 neighbours considered when selecting anchors, reciprocal PCA as the integration method, and default values for all other parameters. UMAPs were run using the sketch integrated data with the first 30 PCA components and projected onto the remaining cells.”

Finally, please note that the entire code is already accessible on Github as mentioned in the code availability section of the manuscript (https://github.com/bdsc-tds/Perriot_Jones_2025). Thus, every step taken for the scRNAseq analysis performed in these experiments, from clustering to statistical analyses, is accessible.

References

1. Simard C, Vogrig A, Joubert B, Muniz-Castrillo S, Picard G, Rogemond V, et al. Clinical spectrum and diagnostic pitfalls of neurologic syndromes with Ri antibodies. *Neurol Neuroimmunol Neuroinflamm*. 2020;7(3).
2. Hebert J, Riche B, Vogrig A, Muniz-Castrillo S, Joubert B, Picard G, et al. Epidemiology of paraneoplastic neurologic syndromes and autoimmune encephalitides in France. *Neurol Neuroimmunol Neuroinflamm*. 2020;7(6).
3. Lutz R, Grunschlager F, Simon M, Awwad MHS, Bauer M, Yousefian S, et al. Multiple myeloma long-term survivors exhibit sustained immune alterations decades after first-line therapy. *Nat Commun*. 2024;15(1):10396.
4. Punyawatthanakool S, Matsuura R, Wongchang T, Katsurada N, Tsuruyama T, Tajima M, et al. Prostaglandin E(2)-EP2/EP4 signaling induces immunosuppression in human cancer by impairing bioenergetics and ribosome biogenesis in immune cells. *Nat Commun*. 2024;15(1):9464.
5. Rodriguez-Sevilla JJ, Ganan-Gomez I, Kumar B, Thongon N, Ma F, Chien KS, et al. Natural killer cells' functional impairment drives the immune escape of pre-malignant clones in early-stage myelodysplastic syndromes. *Nat Commun*. 2025;16(1):3450.
6. Jilek S, Schluep M, Meylan P, Vingerhoets F, Guignard L, Monney A, et al. Strong EBV-specific CD8+ T-cell response in patients with early multiple sclerosis. *Brain*. 2008;131(Pt 7):1712-21.
7. Mathias A, Perriard G, Canales M, Vuilleumier F, Perrotta G, Schluep M, et al. The VZV/IE63-specific T cell response prevents herpes zoster in fingolimod-treated patients. *Neurol Neuroimmunol Neuroinflamm*. 2016;3(2):e209.
8. Frieser D, Pignata A, Khajavi L, Shlesinger D, Gonzalez-Fierro C, Nguyen XH, et al. Tissue-resident CD8(+) T cells drive compartmentalized and chronic autoimmune damage against CNS neurons. *Sci Transl Med*. 2022;14(640):eabl6157.

RESPONSE TO REVIEWER COMMENTS

Reviewer #2 (Remarks to the Author):

Dear authors,

Thank you for carrying out the extra analyses. The new patient-level aggregation and mixed-effects modelling, along with the per-patient plots, fully resolve my concern about pseudoreplication.

To make these points fully transparent to readers, I have just two small wording requests:

1. Add one sentence to the Discussion noting that no genes were significant at the patient level in the pseudobulk/mixed-model analysis (3 controls vs 5 Ri-AIE), most likely because of limited sample size and uneven cell numbers.

2. Results, p. 7, lines 242–252: Where you write that several markers “displayed a significant increase/decrease,” please flag that the significance is at the single-cell level—for example, “(significant at the single-cell level)” after the first occurrence or add a brief footnote stating that significance in this section derives from single-cell tests.

These are purely textual clarifications; no further analyses are needed.

Best regards,
AL

We thank the reviewer for their final appreciation of our work and are pleased to read that our additional analysis fully answered his concerns. As requested, we have added the textual clarifications where needed.

Reviewer #3 (Remarks to the Author):

Although some of the questions raised before still remain in my view and in particular the low number of patients enrolled that might question the relevance of the study, the methodological approach developed is certainly of note and the authors have provided satisfactory answers, although not exhaustive, to the points raised in the previous round of revision.

We thank the reviewer for their appreciation of our work, in particular of the methodology we have developed. We do not contest that the number of patients involved in this study is not high but as explained during the previous round of reviews, it is already a great achievement for this kind of diseases. We heartfully hope the international consortia would develop in the future to enable larger studies on AIE.